

# Long-Wavelength Steric Sea Level and Heat Storage Anomaly Maps by Combining Argo Temperature and Salinity Profiles with Satellite Altimetry and Gravimetry

Don P. Chambers[1] and Sara J. Reinelt[1]

[1] College of Marine Science, University of South Florida, St. Petersburg, FL, USA

*Correspondence to*: Don P. Chambers (donc@usf.edu)

**Abstract.** Argo profiles of temperature/salinity (T/S) at specific times and locations from January 2003 through December 2023 are mapped into monthly maps of steric sea level (SSL), thermosteric sea level (TSL), and Ocean Heat Content (OHC) anomalies at long wavelengths. The mapping uses a monthly satellite reference computed from the difference between

satellite altimetry and gravimetry, so that in periods where there is not sufficient global Argo coverage (generally before 2007), the satellite estimate is used instead of a mean climatology. Longwave mapping is done to reduce large errors introduced by poor sampling of mesoscale eddies by the Argo floats. We demonstrate that on global- and basin-scales, the longwave mapping does not substantially affect calculations of mean SSL, TSL, or OHC changes. Monthly standard error maps from the mapping are also provided. These maps are intended for users interested in understanding global- and basin-

scale sea level budgets, as well as understanding changes in ocean heat uptake.

## 1 Introduction

The Argo program was proposed in 1998 to provide a regular (~10-day) and more spatially dense (~ 1 float per 3°x3° grid) of temperature (T) and salinity (S) profiles of the upper ocean (Argo Steering Team, 1998). Since the initial deployments in

1999 and 2000, floats have continued to be placed into the ocean with regular frequency (Wong et al., 2020). By the late 2000s, almost every ocean region between 65°S and 65°N had at least one float within a 500 km radius returning regular T/S measurements to depths of 2000m (**Fig. 1, Fig. S1**). The main exception is in marginal seas and shallow areas, where there tend to be few or no regular measurements. However, it should be noted that although most of the deep ocean areas now have approximately 1-3 profiles per month within a 300-500 km radius, this is likely to still not be sufficient to accurately

map areas with intense mesoscale signals, as we will discuss shortly.

Numerous studies have demonstrated the usefulness of Argo T/S measurements to understand large-scale ocean circulation (e.g., Davis et al., 2005; Kwon and Riser, 2005; Le Cann et al., 2005; Park et al., 2005; Roemmich et al., 2010) as well as steric sea level and ocean heat storage changes (e.g., Willis et al., 2004; von Schuckmann et al., 2014; Johnson et al., 2016; Chambers et al., 2017; Loeb et al., 2021; Liang et al., 2012). The latter have been primarily based on gridded T/S products

derived from the Argo measurements, which began to be updated at monthly intervals in the late 2000s (e.g., Roemmich and Gilson, 2009), although early work mapped vertically-integrated steric sea level and heat storage anomalies from single profiles (Willis et al., 2004).

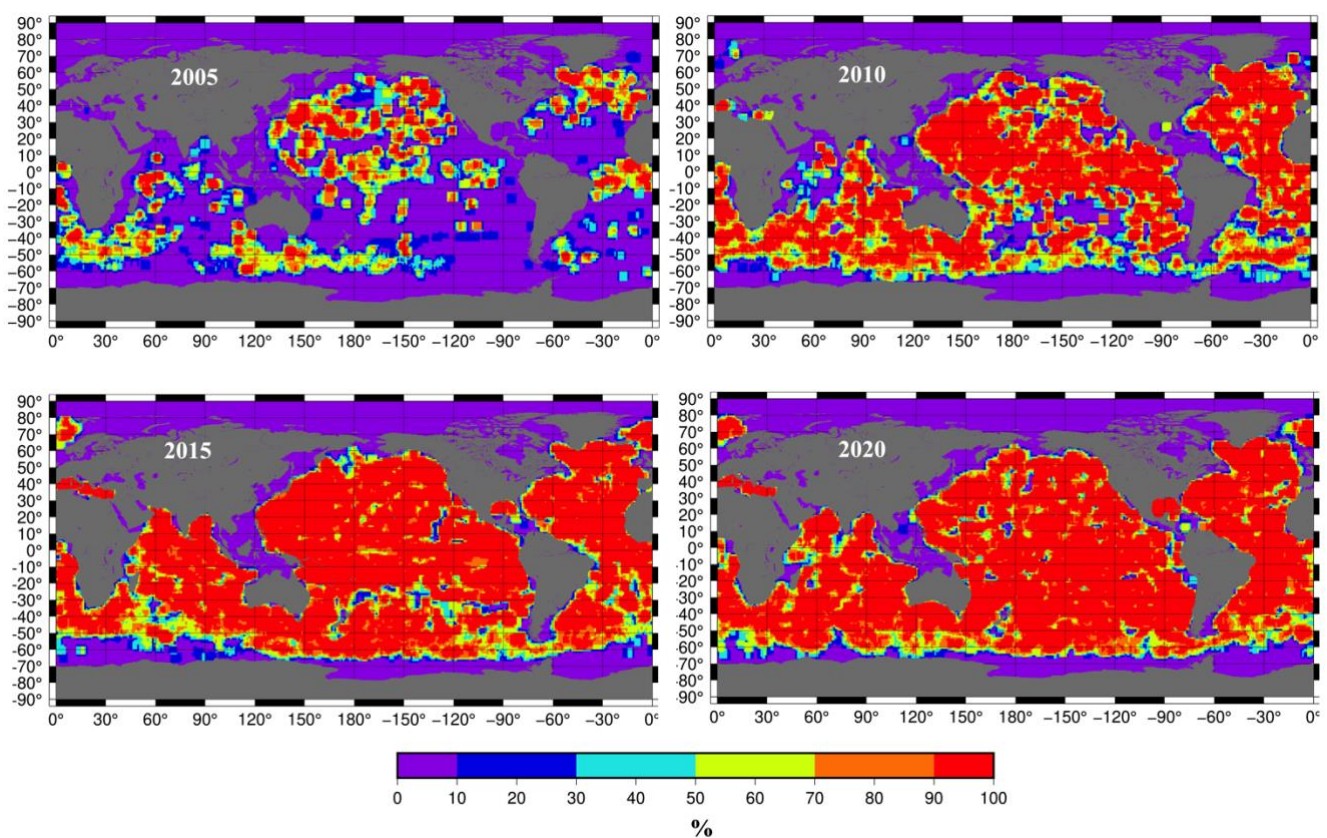

**Figure 1: Percentage of months during the year (2005, 2010,2015,2020) when there is at least 1 Argo profile within 500 km of the**
**center of each 1°x1° grid. Statistics for all years from 2003 until 2023 are shown in Fig. S1-S2.**

Several processing centers around the globe now routinely produce monthly gridded T/S fields, based on different combinations of data, mapping methods, data editing, and the choice of climatology to fill gaps. A non-exhaustive list includes: Scripps Institution of Oceanography (SIO, Roemmich & Gilson, 2009), Barnes Objective Analysis (BOA) from the
Chinese Second Institute of Oceanography (Li et al., 2017), the EN4.2.1 product from the UK Met Office Hadley Centre (Good et al., 2013), the Meteorological Research Institute of Japan (Ishii et al., 2017), and the JAMSTEC center (Hosoda et al., 2008). It has been noted in several papers that while the temperature fields from the products agree well post-2005, there are larger disagreements in salinity fields since 2015 (Liu et al., 2021; 2023). This causes substantial differences in global steric sea level calculations (Blaquez et al., 2018; Chen et al., 2020; Barnoud et al., 2021). Some of these differences may be





caused by use of salinity measurements which have been shown to have drifts or biases in them (Liu et al., 2023; Wong et al., 2023). While suggestions on how to eliminate and correct the problematic floats has been released (Wong et al., 2023), it may or may not have been fully implemented in all the products.

Another method, known as the satellite or geodetic method, can also be used to estimate vertically-integrated steric sea level

and ocean heat content anomalies (Jayne et al., 2003; Hakuba et al., 2021). It relies on differencing the total sea level measured by satellite altimetry and the mass (non-steric) component of sea level measured by the Gravity Recovery and Climate Experiment (GRACE) (e.g., Chambers, 2006) and the subsequent GRACE-Followon (GRACE-FO) missions. For heat storage, the sea level residual must be appropriately scaled to convert to heat storage (Chambers et al., 1997; Jayne et al., 2003; Section 2.3). Besides intrinsic uncertainty differences between Argo, altimetry, and GRACE data, there are

expected to be small differences in the calculations from the two methods, since the geodetic method theoretically accounts for the entire water column, whereas the Argo method can only measure the variations for the upper 2000m. However, deeper ocean warming and sea level contributions derived from full-depth hydrographic sections are expected to be small (e.g., Purkey and Johnson, 2010; Desbruyères et al., 2016), with the largest signals in the Southern Ocean. Numerous studies have compared the two methods on regional and global scales finding good agreement in ocean heat content (von

Schuckmann et al., 2014; Hakuba et al., 2021) and global thermosteric sea level (Blaquez et al., 2018; Barnoud et al., 2021), but with substantial differences in steric sea level after 2015 (Blaquez et al., 2018; Chen et al., 2020; Barnoud et al., 2021), likely due to salinity problems in the gridded Argo T/S products.

One aspect of mapping Argo data that is problematic is how to treat high-variance, mesoscale signals like eddies in mapping

the background T/S state. It is well documented that eddies are distributed around the ocean (e.g., Chelton et al., 2007) and that Argo floats can become trapped in them for months at a time (e.g., Keppler et al., 2024). Eddies can cause significant departures in the local T/S field, and if an Argo float is trapped within one for an extended time, it can potentially bias the T/S profile away from the surrounding state, introducing significant errors into the mapped T/S values away from the true mean value for the grid cell. While there may be several floats within an eddy region every month, there is not sufficient

coverage to completely map the eddy field. Thus, any Argo mapped data will have larger errors in regions of high mesoscale eddy activity, and the errors will depend on the specific Argo sampling, locations of eddies, and the mapping strategy. To our knowledge, the mapping errors in Argo mapped products which would highlight this are not routinely provided.

In this study, we describe a new long-wavelength mapped dataset of steric and thermosteric sea level and heat storage

anomalies, based on a statistical combination of Argo and satellite (altimetry and gravimetry) data at monthly intervals since 2003. We will demonstrate that a longwave mapping method estimates the large-scale steric and heat storage with globally consistent and small errors compared to mapping with eddy variances included in signal covariance – in this case, the Argo sampling causes large errors in western boundary currents and throughout the Southern Ocean. The satellite data will form



the base monthly climatology that Argo data are referenced to, analogous to the method employed by Willis et al. (2004,
2005), although they used only satellite altimetry data. We will demonstrate that before 2007, the use of a monthly-varying
satellite-based reference reduces mapping errors and improves global steric sea level compared to using a mean monthly
climatology (or zero as a reference).

Finally, we also include a map of estimated standard error in the recovered fields, which is derived as part of the optimal
interpolation method. This is rarely distributed with other gridded products and can quickly and easily show where Argo
sampling is too sparse to accurately measure the longwave steric sea level or ocean heat content. Additionally, we also
distribute the specific vertically-integrated Argo profiles used in the mapping, so that non-experts can perform their own
mapping of steric, thermosteric, halosteric, or ocean heat content anomalies using alternate methods without having to
download, quality check, and integrate raw Argo profile data.

Section 2 will describe the specific data and methods utilized, including details on the Argo data editing, integration to steric
sea level and OHC anomalies, and the long-wavelength optimal interpolation and error calculations. Section 3 will analyze
the grids, showing how including eddy variance in the mapping function can introduce large errors in western boundary
currents and the Southern Ocean. We will also demonstrate differences that result from the choice of climatology and that
global and regional averages of the long-wave gridded altimetry data results in nearly identical results to averaging the raw,
unmapped data, thus indicating the maps are capturing the full, longwave signals. Finally, Section 4 will discuss uses for the
data as well as limitations.

## 2 Data and Methods

### 2.1 Argo Data

Profiles of temperature (T) and salinity (S) were downloaded from all Argo autonomous profiling floats for 2003 to 2024
from the Argo Global Data Assembly Center (GDAC) (Argo, 2000; Wong et al., 2020). In addition, gridded mean
climatology T and S values (averaged over 2004-2018) were downloaded from the SIO analysis (Roemmich and Gilson,
2009), in order to compute anomalous steric sea level ($\Delta SSL$), thermosteric sea level ($\Delta TSL$), halosteric sea level ($\Delta HSL$),
and heat storage ($\Delta H$) for each profile. Relevant variables from each float were extracted, including float identification
number, sea water pressure (P), in situ temperature, practical salinity, longitude, latitude, and date. Quality control (QC)
flags associated with data mode, position, and measurements of pressure, temperature, and salinity were also extracted.

Argo profile data were subjected to stringent quality control (QC) steps. Only profiles in delayed mode ("D" flag) were
retained for analysis. Additionally, profiles were retained only if the QC flags for position, date, pressure, salinity, and



temperature were all set to "1," indicating the highest quality measurements (Wong et al., 2023). We should note that flags on some T/S profiles in the online archive have changed since publication of Wong et al. (2023) as QC has improved. In an early download of the profile data in March 2023, we found a small number of profiles with measurements of pressure, temperature, and salinity exceeding predefined extrema or with near-surface pressure measurements erroneously shifted to

deeper positions in the profile and deeper pressures in the surface bins positions. The data downloaded in October 2024 had all these problems corrected or flagged as "bad" properly. Since our goal is to integrate vertically to obtain a value for upper ocean *ΔSSL*, etc, we kept only profiles that were deeper than 750 dbar, had a minimum pressure ≤ 50dbar, and that had more than 50 observations in each profile.

Each Argo profile was matched to the climatology dataset by comparing the longitude and latitude of the Argo profile to the climatology grid, which has a 1° resolution. We did not interpolate, but simply used the value for the grid cell that the Argo profile lay in.  The temperature and salinity values from climatology were vertically-interpolated to match the specific pressure levels of each Argo profile, however.

Practical salinity values were converted to absolute salinity ($S$), and in situ temperatures ($T$) converted to conservative temperatures ($T_c$) (for sea level calculations) and potential temperatures ($T_p$) (for heat calculations), using the Gibbs SeaWater (GSW) Oceanographic Toolbox of TEOS-10 (McDougall and Barker, 2011). Isobaric heat capacity ($c_p$) was computed from absolute in situ temperature and pressure ($P$), while in situ density ($\rho$) was computed from S, $T_c$, and $P$ using the GSW package.


Each Argo profile was then vertically integrated to estimate steric (ΔSSL), thermosteric (ΔTSL), halosteric (ΔHSL) sea level anomalies and heat storage (ΔH) anomalies:

$$\Delta\text{SSL} = -\frac{1}{\rho_0}\int_{-h}^{\eta}[\rho(T_c, S, P, t) - \rho(\overline{T_c}, \bar{S}, P)]dz \tag{1}$$

$$\Delta\text{TSL} = -\frac{1}{\rho_0}\int_{-h}^{\eta}[\rho(T_c, \bar{S}, P, t) - \rho(\overline{T_c}, \bar{S}, P)]dz \tag{2}$$

$$\Delta\text{HSL} = -\frac{1}{\rho_0}\int_{-h}^{\eta}[\rho(\overline{T_c}, S, P, t) - \rho(\overline{T_c}, \bar{S}, P)]dz \tag{3}$$

$$\Delta\text{H} = \rho c_p \int_{-h}^{\eta}[T_p(P, t) - T_p(P)]dz \tag{4}$$

where $\eta$ is sea level, $-h$ is our bottom depth value for the given profile and $\rho_0$ is the reference density of 1027 $kg/m^3$.
Overbars denote climatological values computed from the 2004-2018 Roemmich-Gilson Argo Climatology temperature, salinity, and pressure.





Profiles with ΔSSL >= 2 meters were excluded, as these extreme values were considered erroneous based on the expected range of steric heights. Although we do not directly map halosteric anomalies (since satellite measurements are a poor

reference for halosteric sea level), they are distributed for users who which to map them or to compare the values to those derived from other gridded salinity products, along with all other integrated profile values for all floats used in the mapping (available from Chambers and Reinelt, 2025).

## 2.2 Satellite Data

We utilize along-track, 1-Hz sampled sea surface height anomaly (SSHA) from the Jason-1, Jason-2, Jason-3, and Sentinel-6

nadir altimeter missions, taken from the Integrated Multi-Mission Ocean Altimeter Data for Climate Research (Version 5.1) provided by Beckley et al. (2010; 2022). These data have consistent geophysical corrections and orbits and have had all necessary corrections applied. While gridded multi-mission data are available (e.g., Ducet et al., 2000), we do not use these, as they have already been optimally interpolated accounting for eddy variance in the signal covariance (see Section 2.3). It has been shown that this OI already attenuates longer wavelength signal between 100km and 500km more than 20%

(Ballorata et al., 2019). Thus, using these gridded data would mean reduced signal from some wavelengths that are still present in the original nadir altimetry. Additional smoothing on top of this would attenuate these signals further.

Therefore, it is preferrable to start from the original track observations rather than data that has already had an optimal interpolation scheme applied. As we will demonstrate, the track data from Jason-1, Jason-2, Jason-3, and Sentinel-6 is more

than sufficient to capture the long-wave signal. We do preprocess the raw 1-Hz data by averaging tracks over calendar months and to a preliminary 0.5° grid (noting that many of the grid cells are empty where there are no satellite tracks). This is done primarily to optimize the OI calculations.

We utilize Level-3 Ver. RL06Mv02 GRACE and GRACE-FO (hereafter GRACE/FO) mascons distributed by Jet Propulsion

laboratory (Watkins et al., 2015; Wiese et al., 2019) for our calculations. To be consistent with satellite altimetry, we remove a global mean monthly ocean atmospheric pressure signal from the mascons. This is removed from satellite altimetry as part of the inverted barometer correction but has typically been retained in GRACE/FO data to as it is a signal that occurs in bottom pressure recorders (Chambers and Schröter, 2011). This can be computed directly from the GAD files that are also distributed with the mascons. There are also several large earthquake-related gravity signals in the GRACE/FO mascons that

are not in the satellite altimetry data, from the 2004 Andaman-Sumatra magnitude 9.2 earthquake and the magnitude 9.0 Tōhoku, Japan earthquake in 2011(e.g., Chen et al., 2007; Han et al., 2008; Cambiotti & Sabadini, 2012; Dai et al., 2014). These signals are orders of magnitude larger than the oceanographic mass signals. If we left them in, they would create large, erroneous steric signals in the reference grids. We chose to mask out and not use in our calculations. Thus, when we combine with altimetry to estimate initial guesses at steric sea level, these small areas do not include a mass estimate and simply





approximate SSL as the altimeter measurement. This introduces a much smaller error than including the large solid earth

signal.

 GRACE data is generally complete from January 1 2003 until December 2010 (with only one missing month in June 2003).

Starting in January 2011, however, the GRACE satellites began to suffer battery problems, and the scientific instruments had

to be powered down for 1-2 months at a time (Tapley et al., 2019). This continued until the end of the mission (June 2017)

then there is a 1-year gap before GRACE-FO observations are available in July 2018. There is a 2-month gap in Sept. and

Oct. 2018, then GRACE-FO is continuous. To deal with these gaps, we first estimate a linear trend + annual + semi-annual

sinusoid for each mascon grid cell over the entire record (2003-2023), then evaluate that model at the missing month

midpoint to create a best estimate of the mass value. We then compute the satellite SSL anomaly ($\Delta SSL_{sat}$) as:


$$\Delta SSL_{sat}\,(x,y,t) = \mathrm{SSHA}(x,y,t) - \Delta \mathrm{SL}_{\mathrm{GRACE/FO}}\,(x,y,\,t) \tag{5}$$

where $t$ is time in monthly increments, $x$ is longitude of the nadir track, and $y$ is latitude of the nadir track, at 0.5° increments

along-track. Because $\Delta \mathrm{SL}_{\mathrm{GRACE/FO}}$ is gridded at a lower resolution than 0.5°, the same GRACE/FO value is often used for

subsequent 0.5° cells.

**2.3 Optimal Interpolation Methods**

Optimal Interpolation (OI) is a type of weighted-average of limited observations, except the weights are determined from the

autocovariance of the data and are not arbitrarily defined (such as in a boxcar or Gaussian weighting). If the autocovariance

(and weights) are defined properly, the residuals should be random and reflect the standard error of the optimal value. An

optimal value of $\Delta SSL(x,y,t)$, for example, would be:

$$\langle \Delta SSL\,\rangle_{OI} = \sum_{j=1}^{M} \alpha_j\, \Delta SSL_j = \tilde{\alpha}\widetilde{\Delta SSL}, \tag{6}$$

where $\alpha$ are the weights to be determined from the autocovariance function and the counter $j$ indicates the specific

observation ($M$ total) within some discrete radius of the point where the optimal value is desired. In matrix form (indicated

by $\tilde{\ }$), the weight matrix ($\tilde{\alpha}$) is 1xM, and the observation matrix ($\widetilde{\Delta SSL}$) is Mx1. Time ($t$) can be included in the

autocovariance calculation, but it is normally ignored and all values with a certain period (e.g., a month for our calculations)

are considered identically.

There are several different methods to determine the optimal weights ($\alpha$). We prefer the method described by Wunsch

(2003):



$$\tilde{\alpha} = R_{ij}^{signal}\left[R_{jj}^{signal} + R_{jj}^{error}\right]^{-1}, \tag{6}$$

where $R_{ij}^{signal}$ is a 1x$M$ matrix that contains the **signal** covariance value based on the distance between the grid center ($i$) and the observation locations ($j$) within a window, $R_{jj}^{signal}$ is a $M$x$M$ matrix that contains the **signal** covariance value based on the distance between all observation locations ($j$) within a window, and $R^{error}$ is a $M$x$M$ matrix that contains the **error** covariance value based on the distances between all observations in the window. Because the computational time of the OI is dependent on $M$ (which increases with the search radius), we utilize a search radius of 1500 km. We tested larger windows, and the resulting differences in maps was minimal (errors well below the estimated errors as discussed below), but processing time was often 4-10x slower.

We compute an autocovariance function from the global, ungridded $\Delta SSL_{sat}$ values (Section 2.2) as a function of distance between points or an observation and the center of the grid ($r$). A single autocovariance was computed for all months between 2003 and 2023 – we tested computing month-specific functions, but the differences were minimal, so we used the single covariance function to reduce complexity. We then approximated the covariance values with a continuous function comprised of a Gaussian for short wavelength signals and exponential decay functions for the longwave portion, along with a random component to match the full variance of the data, similar to that done in previous studies (e.g., Willis et al., 2008; Roemmich and Gilson, 2009). Optimal parameters for the roll-off and amplitude parameters were estimated using non-linear least squares based on iterating values of the roll-off parameters over a range of expected values. The covariance function used to calculate the $R$ matrices in Eq. (6) for the satellite SSL is:

$$C^{sat}(r) = 82e^{-\left(\frac{r}{100\,km}\right)^2} + 25e^{-\left(\frac{r}{1675\,km}\right)} \tag{7}$$

with units of cm$^2$, plus an additional 25 cm$^2$ for lag 0 from random variability.

Random error is generally assumed in Equation (6), meaning the $R^{error}$ matrix is diagonal only and the recovered maps will include the short-wavelength variance (albeit, with sampling error). However, there is nothing in formulation that requires that $R^{error}$ only be random. In fact, some of the signal (for instance, the short-wave portion of the covariance) can be treated as a correlated "error" and so, the resulting OI map will reflect the long-wave signal only. This is what we do in our analysis, so that

$$C_{signal}^{sat}(r) = 25e^{-\left(\frac{r}{1675\,km}\right)}$$

$$C_{error}^{sat}(r) = 82e^{-\left(\frac{r}{100\,km}\right)^2} + 25\,cm^2\,(random). \tag{8}$$



Using this formulation means that the $R^{error}$ matrix is no longer diagonal but has off-diagonal terms. Note that because the

number of observations ($M$) are not necessarily uniform in number for all grid cells (e.g., from missing tracks or gaps), the sizes of the $R$ matrices in Equation (6) will vary from grid cell to grid cell and time to time, so they have to be computed independently for each grid cell and month.

One benefit of an optimal interpolation is that a standard error map ($\sigma_i^2$) can also be directly calculated from the $R$ matrices in Equation (6) (Wunsch, 2003):

$$\sigma_i^2 = \sigma_{exp}^2 - R_{ij}^{signal}\left[R_{jj}^{signal} + R_{jj}^{error}\right]^{-1} R_{ij}^{signal\,T}, \tag{9}$$

where $\sigma_{exp}^2$ is the expected variance of the maps (based on the global covariance) and the matrix calculation provides the actual variance based on the distances between data and number of observations. Thus, the difference reflects the mapping error based primarily on the sampling of the data. For low number of observations within the window, the error will be

higher than for larger numbers. For our mapping of the satellite SSL, we use $\sigma_{exp}^2 = 25$ cm$^2$, the value used in the long-wave signal covariance amplitude.

For data that may have large spatial gaps (e.g., early Argo) with many areas having M < *nmin* observations within the monthly window, a climatology is commonly used (e.g., Willis et al., 2004; Roemmich and Gilson, 2009) to fill the gaps.

For our calculations, we use *nmin* = 10. Typically, the climatology is removed from the data when mapping, so that residuals to the reference surface are mapped, then the reference is restored. In our case, that would be:

$$\Delta SSL_{combined}(\phi, \lambda, t) = \langle \Delta SSL_{Argo} - \Delta SSL_{sat\_oi}\rangle_{OI} + \Delta SSL_{sat\_oi}, \tag{10}$$

where the subscript *sat_oi* indicates the monthly mapped version of $\Delta SSL_{sat}$ in Eq. (5) and $\Delta SSL_{Argo}$ are the profiles in Eq. (1). Because using a reference removes some of the variance, the covariance functions should be modified to reflect this. For our long-wave mapping of data we use:

$$C_{signal}^{Argo}(r) = 18e^{-\left(\frac{r}{1675\,km}\right)}$$


$$C_{error}^{Argo}(r) = 60e^{-\left(\frac{r}{100\,km}\right)^2} + 4\,cm^2\,(random) \tag{11}$$



with $\sigma_{exp}^2 = 18$ cm$^2$ (Eq. (9)), which is based on the autocovariance structure of Argo profile SSL after removing the long-wave mapped satellite SSL.


However, before combining the satellite and Argo data in the OI scheme, one must account for differences in the reference period used for the satellite altimetry, GRACE/FO, and Argo data. GRACE/FO is referenced to a 2005-2010 mean surface, the altimetry to a mean surface for approximately 1993-2018, and Argo 2004-2018. If one does not reference the $\Delta SSL_{sat}$ reference grids to a comparable time-period as Argo, this would potentially cause biases from 2003 to about 2008 when

Argo data is not complete (**Fig. 1**, **Fig. S1**). Removing a mean surface for 2004-2018 from $\Delta SSL_{sat}$ may also not completely solve the problem, since: 1) there are gaps in GRACE/FO data at this time, and 2) the Argo climatology in some areas is likely biased more toward 2008-2018 because of significant gaps in coverage before this period. Therefore, we performed a multi-step process to address the problem:

1. The original $\Delta SSL_{sat}$ along-track data was mapped to create $\Delta SSL_{sat\_oi}(x,y,t)$.
2. The Argo profile data ($\Delta SSL_{\mathrm{Argo}}$) was mapped without using a reference to create $\Delta SSL_{Argo\_oi}(x,y,t)$.
3. Monthly differences were computed ($\Delta resid(x,y,t) = \Delta SSL_{sat\_oi}(x,y,t) - \Delta SSL_{Argo\_oi}(x,y,t)$), then averaged over 2008-2016 to create $bias(x,y)$. This provides a map of the mean difference between the satellite maps and the Argo maps for the period 2008-2018, when Argo coverage is highest.
4. A new set of satellite maps is created ($\Delta SSL_{sat\_oi\_corr}(x,y,t) = \Delta SSL_{sat\_oi}(x,y,t) - bias(x,y)$). These maps are then used as the *a priori* first guess in the OI and the Argo profiles are referenced to them (Eq. (10)-(11)) in the OI.

For mapping thermosteric sea level anomalies from Argo, the corrected satellite SSL maps are used along with the

thermosteric profiles from Argo (Eq. (2)). This is a reasonable proxy in the midlatitudes (Chambers et al., 1997; Jayne et al., 2003), but less reasonable at higher latitudes because of larger salinity fluctuations there. This should be considered if using the thermosteric maps before more complete coverage of Argo in ~2008.

For heat storage anomalies, we computed a scale factor ($\gamma$) from all the Argo thermosteric and heat storage anomaly profiles

based on a least squares fit of all profiles over time in each 1° grid cell (**Fig. 2**) so that

$$\Delta H(x,y,t) \sim \gamma(x,y)*\Delta TSL(x,y,t). \tag{12}$$

We estimate the heat storage anomalies from the satellite SSL maps by scaling them by the value of $\gamma$ in the appropriate grid

cell, then use this as the reference to map the Argo profiles of heat storage anomalies for the optimal interpolation, noting



that the amplitudes in Eq. (11) have to be scaled to account for the conversion from cm of SSL to heat storage (in units of $10^7$ J m$^{-2}$).

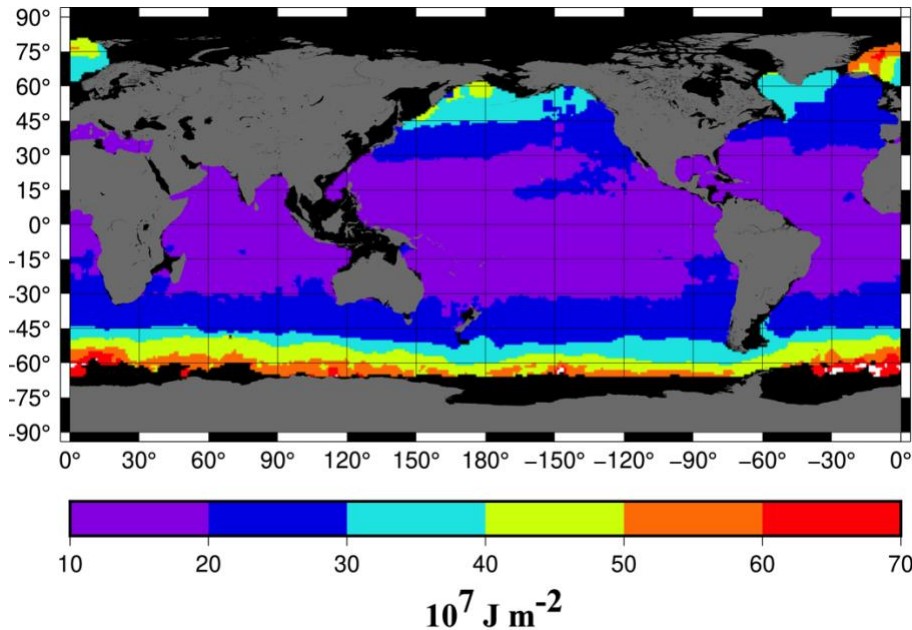

Figure 2: Estimated scale factor to convert sea level anomalies (in cm) to heat storage anomalies (in $10^7$ J m$^{-2}$).


Finally, we mask the mapped data to remove any estimates in marginal seas or shallow water and above ±65° latitude to be consistent with the SIO no data mask. While we sometimes have sufficient data to find a value in marginal seas, errors are quite large (or are based primarily on the satellite reference) and the OI value mainly depends on observations outside the marginal sea. So, like the SIO product, we choose to ignore them.


### 3 Analysis of Mapped Data

We first demonstrate the effect of sampling eddies with limited Argo floats. To do this, we performed a sampling experiment using the altimetry track data. First, the nearest along-track altimetry points in each month to each Argo float location and date are interpolated using bilinear spatial and linear temporal interpolation. These sampled values are then mapped using the two different covariance functions and the standard deviation of the mapped values computed (**Fig. 3**). Standard deviations of the "eddy-resolution" Argo maps differ from the altimetry maps by over 12 cm in all regions of high mesoscale activity (**Fig. 3a**) and between 4 and 8 cm in many other ocean regions. This is due solely to the sampling of the Argo floats: i.e., the sampling, even in later years, is insufficient of fully resolving the short-scale (<100km) variance in the sea level as


opposed to the many more altimeter observations. Conversely, differences in the longwave maps peak at about 3-4 cm

standard deviation in mesoscale regions, with differences of less than 2 cm over most of the ocean (**Fig. 3b**). The only exception is in areas with limited Argo floats and these areas are masked in the final products.

The sampling error is not perfectly random. To demonstrate this, the sampling error from the experiment (sample – full) is plotted for two grid cells in **Fig. 4**, one in the Kuroshio extension (150.5°E, 40.5°E) and one in the Agulhas Retroreflection

zone (30.5°E, 40.5°S). Biases as large as 20 to 40 cm can persist for upwards of 6 months, resulting in significant non-zero trends (± 10 mm yr$^{-1}$).

The longwave mapping does not significantly alter the global mean sea level estimate (**Fig. 5**) or basin-scale averages (**Table 1**). Correlations exceed 0.9 (p < 0.001) in every case, standard deviations range from a low of 1 mm for the global average to

3 mm for regional averages. Trend errors range from 0.08 mm yr$^{-1}$ for the global average, to about 0.3 mm yr$^{-1}$ over basins. Thus, the longwave maps are sufficiently accurate to resolve global and regional sea level budgets.

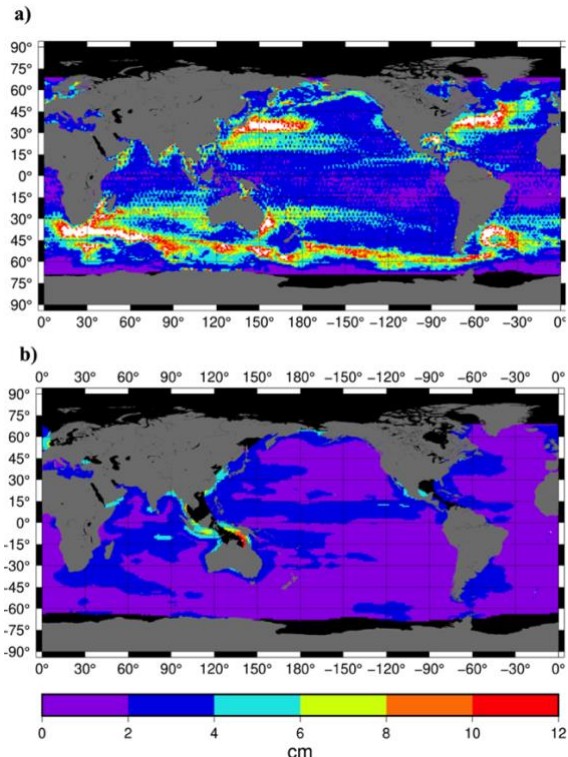

**Figure 3: Standard deviation of Argo sampling experiments, where altimetry interpolated to Argo profile locations and times is**

**mapped and compared to maps from the full altimetry. a) Eddy covariance as signal and b) eddy covariance as error (longwave signal).**



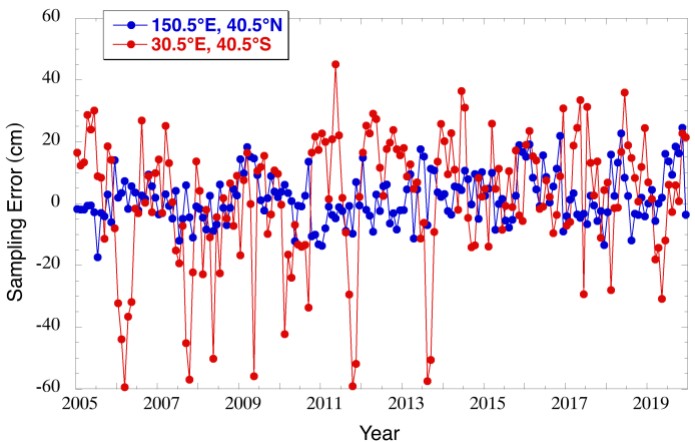

**Figure 4: Argo sampling error at two regions of high mesoscale eddy activity.**


**Table 1: Statistics for basin-averages (2005-2019). Difference are eddy OI – longwave OI.**

| Basin | Trend Difference (mm yr$^{-1}$) | Correlation | Std. Dev. Of Difference (mm) |
|---|---|---|---|
| Global | 0.08 | 0.99 | 1.0 |
| N. Pacific | 0.09 | 0.99 | 1.4 |
| N. Atlantic | -0.04 | 0.99 | 2.6 |
| Tropical Pacific | 0.16 | 0.99 | 1.1 |
| Tropical Atlantic | 0.15 | 0.99 | 1.8 |
| Indian Ocean | 0.27 | 0.99 | 1.6 |
| S. Pacific | -0.37 | 0.99 | 1.9 |
| S. Atlantic | 0.33 | 0.99 | 2.8 |
| Southern Ocean | 0.33 | 0.98 | 2.9 |






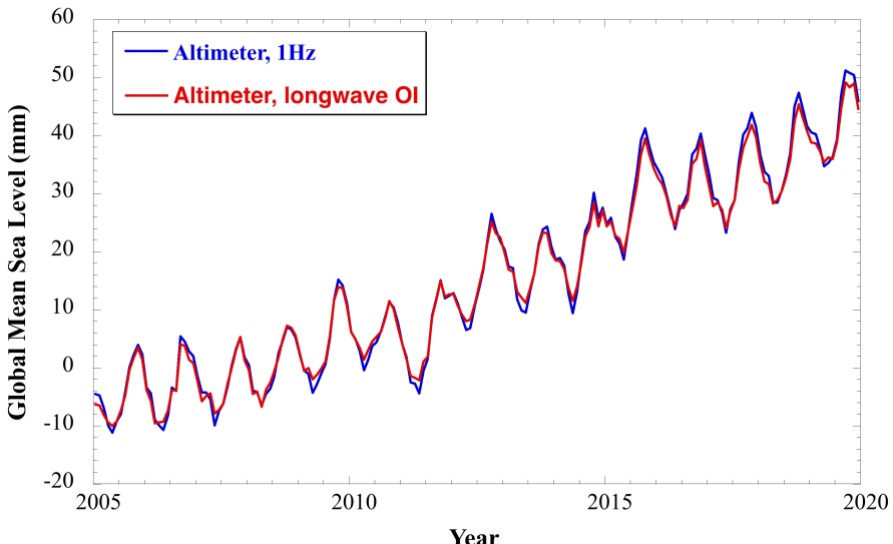

**Figure 5: Global mean sea level from raw, 1-Hz altimetry (blue) and from longwave OI maps (red).**

The choice of climatology has a noticeable effect on recovered Argo SSL maps in the early part of the record, as others have commented on previously (e.g., Lyman and Johnson, 2008). We have performed three different mapping experiments of

Argo SSL to demonstrate this effect on global mean SSL (GMSSL) calculations: 1) using no climatology, so that areas without sufficient Argo profiles are set to no data, 2) using a mean climatology based on Argo mapping only from 2008-2018, and 3) using the monthly satellite estimates (the final product). The results (**Fig. 6**) show that using no climatology and having gaps in coverage significantly impacts GMSSL until at least 2009, with the largest differences occurring before 2008. Even using a mean climatology to fill gaps creates significant biases (~ 5mm) with the satellite reference before 2007,

with smaller (< 2 mm) biases up until at least 2011. While it has been previously accepted that Argo data mapped with a climatology reference is sufficient for studying sea level budgets back to 2005 (e.g., Chambers et al., 2017; Blaquez et al., 2018; Chen et al., 2020; Barnoud et al., 2021), this experiment suggests interannual changes in SSL before ~2008 are sufficiently different from the climatology and that gaps in Argo coverage are still large enough that it can lead to systematic errors in GMSSL. The mean trend from 2005-2024 increases from $1.04 \pm 0.04$ mm yr$^{-1}$ for the climatology reference to $1.17$

$\pm 0.04$ mm yr$^{-1}$ using the satellite reference (error at 90% confidence interval). From 2011 to 2024, when the Argo coverage is more complete and the reference has minimal effect, the trend difference is only $0.08 \pm 0.07$ mm yr$^{-1}$ (90% confidence interval). Trend uncertainty accounts for reduction in degrees of freedom due to non-random residuals as described in Chambers et al. (2017).



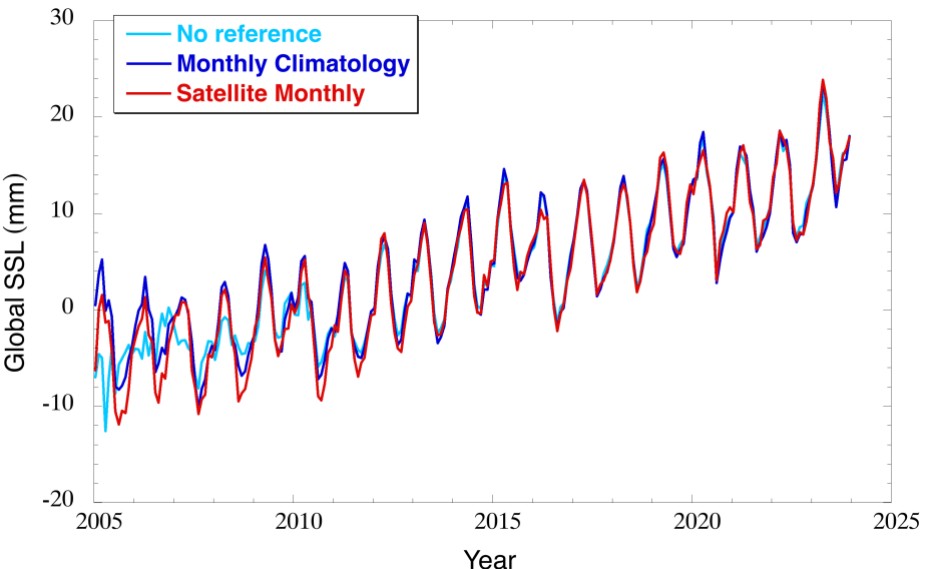

**Figure 6: Global mean steric sea level from reference SSL experiments. (cyan) No reference, so grids without sufficient Argo profiles for mapping defaults to no data, (blue) a mean monthly climatology based on Argo-only maps from 2008-2018, and (red) the released product, utilizing altimetry-GRACE/FO monthly maps.**

As discussed in Section 2.3, error maps are provided for each monthly map of SSL, TSL, and OHC anomalies. It should be noted these represent standard errors based on the quantity of data used in the mapping and how well the mapped data represents the expected longwave covariance; they cannot explain the full uncertainty, such as arising from sampling errors (e.g., **Fig. 3b**) or systematic errors (such as biases or drifts). However, they are useful for seeing where the satellite reference data are used to fill gaps and where the maps are mostly determined by Argo data. Because more data are used in the satellite mapping, the errors are considerably lower than where Argo data is used to update the reference (**Fig. 7**). Note that in early years (2003-2005 especially), the standard errors often have a "bullseye" pattern, with lower errors in the center and higher values increasing with radius (see 2003 and 2005 in **Fig. 7**). This reflects lower errors near a cluster of Argo floats, and larger errors on grids that are within the radius of the OI mapping but with no (or few) Argo floats in them during that month. Oftentimes, there is an abrupt transition from high errors to low errors where there are no Argo floats and so the mapping defaults to the satellite reference.

The satellite data tend to dominate the mapping from 2003 to 2005 but are still the primary source in some areas of the South Pacific as late as 2008 and 2009 (**Fig. 7**; see also Figs. S1 and S2). By 2010, most of the ocean has sufficient Argo coverage each month that maps are computed fully from Argo data and the reference does not matter, so the error maps reflect the Argo mapping error. TSL errors from the satellite data are identical to the SSL errors from satellites (as the same satellite reference maps were used for both TSL and SSL), but the Argo errors differ, since TSL is directly computed from Argo





profiles. OHC errors are merely the TSL errors scaled to the appropriate units.

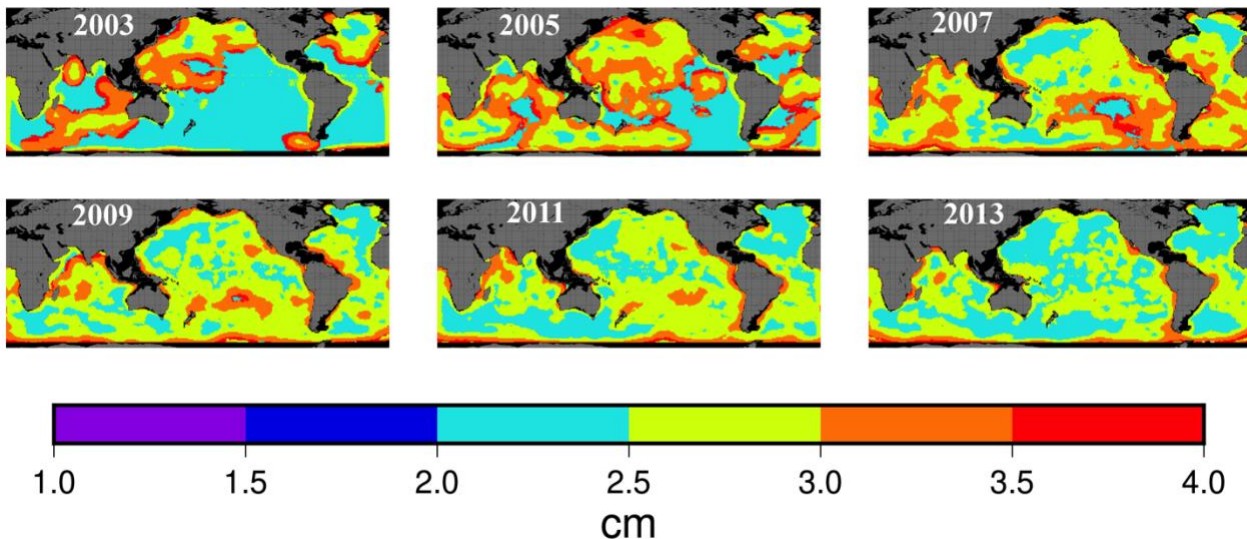

**Figure 7: Standard error for SSL in June for years indicated.**

It is beyond the scope of this manuscript to compare our results to every gridded Argo product available. However, we do compare the global mean steric (**Fig. 8**) and thermosteric sea level (**Fig 9**) computed from our product (designated USF for University of South Florida) with that computed from the SIO gridded T/S data. The SIO product has been shown to have

smaller salinity errors than other available products (Blaquez et al., 2018; Barnoud et al., 2021; Liu et al., 2023). Thus, the comparison provides some insight into how our processing may inform sea level budget studies. We also show a global SSL series computed by differencing the altimetry track data (from Beckley et al., 2022) and GRACE/FO (from the JPL Release RL06Mv02 mascons) – the same data used for the longwave gridded reference grids – but here we do no additional mapping, just perform a global area-weighted average of the original resolution data. Several minor corrections are applied

to remove signals that the Argo data do not observe (see Blaquez et al., 2018 for why these are necessary): 1) a correction for glacial isostatic adjustment is added to the altimetry (Nerem et al., 2010; value used = 0.3 mm yr$^{-1}$), 2) a correction for Jason-3 radiometer drift is added to the Jason-3 altimetry as the product we used did not include this at the time of processing (Brown et al., 2023), and 3) an estimate of the mean global steric sea level rate below 2000 m (0.11 mm yr$^{-1}$) from Purkey and Johnson (2010) was removed to align with the 2000 m max depth used in the Argo profiles.




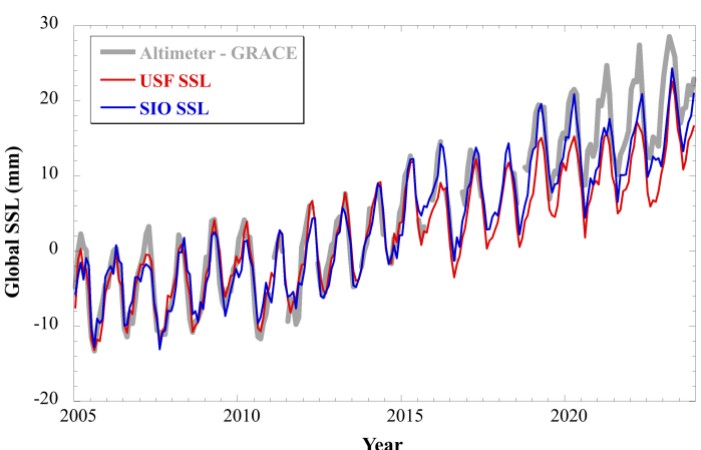

**Figure 8: Global mean steric sea level from altimetry-GRACE (gray), USF mapped SSL (red), and SIO mapped T/S grids (blue).**

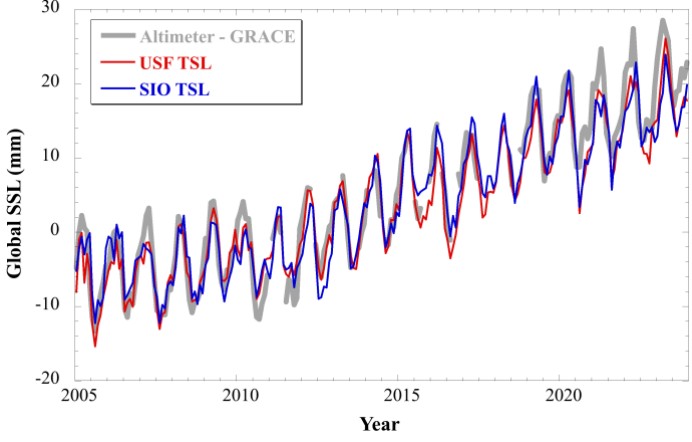

**Figure 9: Global mean steric sea level from altimetry-GRACE (gray), and thermosteric from USF (red), and SIO (blue).**

As noted in multiple studies (Blaquez et al., 2018; Chen et al., 2020; Barnoud et al., 2021), there is a significant misclosure between the satellite estimate and the Argo GSSL estimate after 2019-2020 for both the SIO and USF products (**Fig. 8**). The SIO estimate matches altimetry well in 2019 and the first half of 2020, but then falls off rapidly, while the USF estimate is significantly lower starting in 2018. However, the USF and SIO estimates generally agree after 2020.5, both being significantly lower than the satellite estimate.


From this analysis, we conclude that GSSL computed from our long-wave steric sea level maps is in general agreement with that derived from the currently available SIO T/S maps, except for two notable occasions: in 2015 and between 2019-2020.5. The difference in 2015 is interesting, as the USF estimate agrees better with the satellite estimate at the trough (SIO higher), but not at the peak (USF lower). Moreover, when global thermosteric sea level (GTSL) is compared instead (**Fig. 9**), there is

no change in behavior during 2015, but USF and SIO GTSL agree well for all other periods. This suggests that the behavior





in 2019-2020.5 is due to unresolved salinity errors that are not flagged (and so get into the USF processing) but that SIO has edited out. However, the differences in 2015 can't be explained by salinity errors. We experimented with an additional 3-sigma editing of the T/S profiles (based on difference with the climatological profiles); while that removed a small number of profiles, it did not significantly affect the recovered maps in those periods, or GSSL/TSSL. Because of this, we selected

not to use a 3-sigma editing

Unfortunately, because the exact retained floats and profiles used in the SIO mapping are not provided, we cannot test if our analysis would change if we used the same profiles in our mapping. We suggest it would be beneficial for all Argo data centers producing statistically mapped data to provide a list of the exact float numbers and profile dates used in their analysis

so that mapping experiments with the same raw data can be conducted by other groups. For this reason, we distribute the float numbers and dates that we utilize in our mapping.

A common use of OHC anomalies is to calculate ocean heat uptake (OHU), which accounts for upwards of 90% of the global earth energy imbalance associated with global warming (e.g., Hakuba et al., 2021; Loeb et al., 2021). OHU is the

time-derivative of the global average of OHC anomalies (in W m$^{-2}$), after scaling to account for the smaller surface area of the ocean compared to the total earth surface area (Hakuba et al., 2021). Here, we show that the OHU computed from our OHC grids matches that computed from the SIO T/S grids within estimated errors, except for the periods that include 2015 (**Figure 10**). OHU was computed by estimating running 2-year trends (along with annual and semiannual sinusoids) from the global average OHC (and converting J m$^{-2}$ yr$^{-1}$ to W m$^{-2}$) – the time stamp used is the middle of each 2-year window and a

one-month step was used.

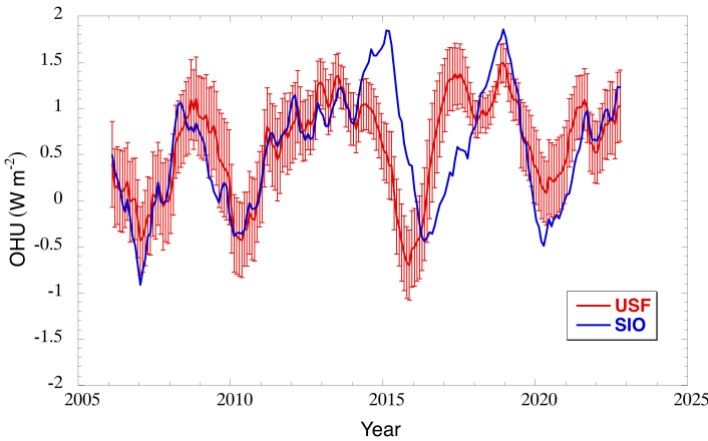

**Figure 10: Ocean heat uptake from USF OHC anomaly grids (red), and SIO T/S grids (blue). Uncertainty is the 90 % confidence interval based on least squares fit, and accounts for reduction serial correlation in residuals as described by Chambers et al.**

**(2017). Uncertain is only shown for the USF calculation, but the SIO uncertainty is similar in magnitude.**



The departure between our estimate and that of SIO between 2014 and 2017 is due to the significant difference in OHC in 2015, already noted in the TSL discussion (**Figure 9**). Even with the interannual difference between 2014 and 2017, the overall means (2005 and 2023) agree within the standard error (USF: $0.58 \pm 0.18$ W m$^{-2}$; SIO: $0.54 \pm 0.21$ W m$^{-2}$) and are in
the range reported by others (e.g., Hakuba et al., 2021; Loeb et al., 2021).

**4 Conclusions**

We have described a new set of steric and thermosteric sea level anomaly gridded data, at monthly resolution between 2003 and 2023. Additionally, ocean heat content anomalies are also available. Maps are based on using a satellite estimate (altimetry – GRACE/FO) as a starting guess, then updating with values derived from Argo profiles. The data have been
intentionally mapped to retain only the longwave portion of the covariance, to reduce sampling errors from limited Argo observations in regions of high mesoscale activity; limited Argo profiles (especially from floats within an eddy) may lead to biased estimates in individual grid cells if the short variance signal is mapped. As we have demonstrated, there is no significant difference between the longwave or shortwave mapped data when averaged over ocean basins and globally.

These data complement mapped data from other sources (e.g., SIO, the Chinese Second Institute of Oceanography, EN4.2.1 the Meteorological Research Institute of Japan, JAMSTEC), but provide useful benefits for scientists working on sea level budget and ocean heat uptake studies:

1) Data are provided in integrated values necessary for these studies, not in T/S at depth that require conversion and
integration.

2) The monthly satellite reference likely improves estimates during 2003 to ~2008 when Argo sampling still has substantial gaps (e.g., **Figure 5**).

3) Error maps are provided for SSL, TSL, and OHC for each month. Such error maps are not routinely provided by other processing centers.

4) The integrated SSL, TSL, halosteric sea level, and OHC anomalies for each float profile that met the flagging and editing criteria is also distributed. This allows users to see the locations and time of the exact floats used in the analysis, which may
be of use for understanding differences between other data center mapping strategies (e.g., the difference between our grids and those of SIO in 2015). Again, this information is not available from other centers that we have been able to find.



There are limitations to the data. Because these are mapped over long wavelengths, they will not capture small-scale variations in sea level or OHC. While this was done to improve the recovery of unbiased values of the background state in 480 areas of high mesoscale activity, it will also reduce signal in other areas, such as in the tropics where there can be signals from shifts in the zonal equatorial currents. They likely will also underestimate El Niño variations in the eastern and western Pacific. Users focusing on these specific regions should take this under consideration.

## Data Availability

All data utilized in this project is publicly available. GRACE data can be downloaded from https://podaac.jpl.nasa.gov/dataset/TELLUS_GRAC-GRFO_MASCON_CRI_GRID_RL06_V2. Altimetry data can be downloaded from https://podaac.jpl.nasa.gov/dataset/MERGED_TP_J1_OSTM_OST_ALL_V51. Argo data were collected and made freely available by the International Argo Program and the national programs that contribute to 490 it. (https://argo.ucsd.edu, https://www.ocean-ops.org). The Argo Program is part of the Global Ocean Observing System. Final mapped data are available at https://doi.org/10.17632/dsjkkhvywr.1 (Chambers and Reinelt, 2025).

## Author Contributions

SJR downloaded and processed the Argo data and wrote the section on Argo processing in the manuscript. She also 495 processed the final data into netCDF files. DPC processed the satellite data, performed the optimal interpolation, conducted the majority of the analysis, and wrote the manuscript.

## Acknowledgements

This research was carried out under grant number 80NSSC23K0353 from the NASA Physical Oceanography program.

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
