# Peer review of "Long-Wavelength Steric Sea Level and Heat Storage Anomaly Maps by Combining Argo Temperature and Salinity Profiles with Satellite Altimetry and Gravimetry"

_Earth System Science Data, 2025_

## Author Comment (AC1)

We would like to thank **Reviewer # 1** for their very thorough review of our manuscript and thoughtful suggestions. Below, we respond to the more significant suggestions, in the order they appear. The reviewer's comment is in italics, while our **Response** follows.

In this paper, the authors use temperature and salinity data from Argo floats (2003–2023) to produce monthly maps of steric sea level (SSL), thermosteric sea level (TSL), and ocean heat content (OHC) anomalies, focusing on long-wavelength signals. A satellite-based reference, derived from the combination of satellite altimetry and space gravimetry, is used as a first a priori field and helps fill data gaps in Argo coverage, particularly before 2007. A long-wavelength mapping approach is applied to reduce errors in the monthly maps caused by the sparse sampling of mesoscale variability by Argo. The authors claim that the resulting dataset, including associated error maps, enables analysis of sea level and ocean heat variations on both global and basin scales.

**Major comments:**

This study addresses the important issue of SSL, TSL, and OHC variability at regional and global scales—key components of the global and regional sea level budget. Continuous and accurate estimates of SSL, TSL, and OHC are essential for closing the sea level budget and understanding sea level variability. However, precise estimates are limited by the number of Argo floats, which sparsely sample the high-variance mesoscale variability induced by eddies and fronts, introducing significant regional errors in TSL and SSL relative to the true mean state. The authors partly overcome this limitation by using a long wave length optimal smoother with a background climatology (a priori estimate) based on a combination of satellite altimetry and gravimetry, and correlation coefficients derived from satellite altimetry data, thereby damping the noise induced by mesoscale variability. As such, this study is relevant to the climate science community.

The use of satellite altimetry to derive the correlation coefficients for the smoother and as a firstguess field is not new—it has been previously employed by Willis et al. (2003 JGR, 2004 JGR ocean) and further by Lyman and Johnson (2008 JCLI, 2023 JAOT) The novelty of the present work lies in combining satellite altimetry with GRACE data instead of relying solely on altimetry. This addition likely has a limited (possibly insignificant) impact on the calculation of correlation coefficients, since GRACE cannot resolve mesoscale variability. However, it may improve the final solution by providing a first-guess field more consistent with Argo data. Unfortunately, this potential benefit is not analyzed in the paper: the authors do not compare solutions with and without gravimetry data to assess any improvement. Another new aspect of the study is the production of maps of interpolation-induced long-wavelength errors. Although these "error maps" do not represent total uncertainty, they are valuable as they indicate where and when satellite reference data are used to fill gaps and where Argo data dominate the reconstruction. This is a welcome addition, as most processing centers do not provide such information. Aside from these aspects, I do not identify significant novelty compared to previous literature. The authors claim that using altimetry in the optimal interpolation improves SSL, TSL, and OHC estimates, particularly before 2008. This is true, but it is already well established (e.g., Lyman and Johnson 2008).

**Response**: We acknowledge that the novelty in our approach is limited and is primarily from the use of GRACE in addition to altimetry and the long-wave focused OI to reduce eddy sampling error. We have added new text in the Introduction (around line 75) to discuss previous mappings of Argo using only altimetry, and studies that have argued for improvement in GRACE is used (e.g., Jayne et al., 2003).

New text in Introduction (new text is in bold): "In this study, we describe a new long-wavelength mapped dataset of steric and thermosteric sea level and heat storage anomalies for the upper 2000m of the ocean, based on a statistical combination of Argo and satellite (altimetry and gravimetry) data at monthly intervals since 2003. While previous studies have mapped Argo data using a reference based solely on satellite altimetry (Willis et al., 2004, 2005; Lyman and Johnson, 2008, 2023), Jayne et al. (2003) suggested from a model experiment that using a combination of altimetry and satellite gravimetry would improve mapped steric sea level and ocean heat content in high latitudes. While such a combination with real GRACE/FO observations has been used in global comparisons (Blaquez et al., 2018; Chen et al., 2020; Barnoud et al., 2021) to compare to already mapped Argo data, it has not been fully tested as a reference in mapping Argo data. As suggested by the limited Argo coverage before 2007 (Figure 1 and Figure S1), use of altimetry-GRACE/FO as a reference to fill gaps may have a significant impact before 2007 in mapping Argo data."

Additionally, we have added text in Section 3 to demonstrate that starting from altimetry-GRACE maps results in a smaller update when combining with Argo than using only altimetry, and this will have an impact in the pre-2007 grids when the reference is important for filling in gaps. We have added a new Figure 3 showing standard deviation between Argo-only maps and altimetry-GRACE/FO and altimetry-only computed over 2010-2023 when Argo sampling is sufficient for mapping without a reference. The results clearly show altimetry-GRACE/FO agrees better with Argo in all regions than altimetry. We have added discussion on this between **lines 336 and 347** of the revised manuscript:

New text: "To demonstrate the relative benefit of using monthly altimetry-GRACE/FO maps as a climatology in the early record, we compare those to maps gridded with only Argo data post 2010 when there is sufficient data to calculate maps with no reference climatology (Fig. 3a). The Argo-only grids are also compared to longwave mapped altimetry grids over the same period (Fig 3b). It is clear that the altimetry-GRACE/FO maps agree better with Argo-only mapping than only altimetry at most areas outside western boundary currents and the Agulhas Retroreflection, indicating that the altimetry-GRACE/FO maps we utilize as a reference are better for filling gaps in Argo data (especially prior to 2007-2008) over using just altimetry as a reference. This is because GRACE/FO removes large non-steric bottom ocean mass variations in the Southern Ocean and North Pacific (e.g., Jayne et al., 2003) as well as the non-steric global ocean mass variations that are present in altimetry data (e.g., Chambers et al., 2017; Chen et al., 2020). Overall, reduction in error before 2007-2008 where the maps default to the reference is expected to be 2-3 cm in the Southern Ocean and about 1 cm in the rest of the ocean north of 45°S."

The general approach of interpolating Argo profiles with satellite data to estimate SSL, TSL, and OHC used in this work, is standard and reasonable. However, in this work, it presents several important limitations that need to be addressed for the results to be convincing and fully comprehensible.

The methodology is not clearly explained.

a) The authors provide "long-wavelength" OHC, SSL, and TSL estimates down to 2000 m depth, but this depth limitation is never clearly stated in the abstract or the main text. It should be explicitly mentioned, ideally in the title.

**Response**: We have added the limitation in the abstract and throughout the text in the revision. Although the title is already quite long, we have added it there as well.

New Title: "Long-Wavelength Steric Sea Level and Heat Storage Anomaly Maps to 2000m by Combining Argo Temperature and Salinity Profiles with Satellite Altimetry and Gravimetry"

b) The term "long wavelength," which is central to the approach, is never quantitatively defined. While Equation 8 provides some indication of the targeted scales, the concept is not clearly explained or discussed in the abstract or main text. There is also no discussion of whether these intended long wavelengths are effectively resolved in the final estimates.

**Response**: Apologies. We assumed this would be clear from the equations used for the OI. We have added text after Equation (7) to be explicit about what we mean by "longwave" and "shortwave."

"Throughout the remainder of this paper, when we say "longwave" or "long wavelength" we mean the portion of SSL that has a covariance function equal to  $25e^{-\left(\frac{r}{1675\,km}\right)}$ , i.e., an exponential decay with a roll-off of 1675 km and max covariance of 25 cm². By "shortwave" or "short wavelength" we mean the portion of SSL that has a covariance function equal to  $82e^{-\left(\frac{r}{100\,km}\right)^2}$ , i.e., a Gaussian with a roll-off of 100 km and max covariance of 82 cm². By "eddy resolution" we mean the full covariance structure of Equation (7). A spectral analysis of the longwave maps compared to the original, unsmoothed data indicates the longwave

We have also added another line to equation (7) to make this separation abundantly clear in the equation:

maps keep nearly 100% of the power for wavelengths longer than 1500 km."

$$C^{sat}(r) = 82e^{-\left(\frac{r}{100 \text{ km}}\right)^2} + 25e^{-\left(\frac{r}{1675 \text{ km}}\right)} , \qquad (7)$$

$$C^{sat}(r) = C^{sat}_{short}(r) + C^{sat}_{long}(r).$$

As far as testing whether these scales are resolved, we have computed the autocovariance of the final mapped data and verified they are the same as assumed in the OI mapping. We have added a statement on this at the beginning of Section 3 in the revised manuscript:

"To verify that the mapping captures the longwave signal, we calculated the mean autocovariance of all the SSL maps and verified it matches the longwave covariance structure used in the OI mapping (Equation (7))."

c) The overall workflow of the method is unclear. It took several readings to infer that the optimal interpolation (OI) scheme is applied multiple times—first to satellite data, then to Argo profile anomalies relative to the satellite field. While the equations are clear, the narrative is not. A schematic diagram of the workflow would greatly aid comprehension.

**Response**: Apologies. We felt it was clear enough in the text. We have added a schematic flowchart to show the process and will include it in Supplemental Material (as the number of Figures already included in the main text is quite high).

d) Some key assumptions underlying the OI approach are not explicitly stated or discussed (see, for instance, my detailed comment on Equation 8 regarding the assumption that the signal is uncorrelated with the errors).

**Response**: Apologies for not detailing this specifically, as we thought it was clear that only random error was assumed, indicating there was no correlated error assumed, as it is unknown. We have now explicitly stated this.

New text added around at end of discussion for Equation (6).

"Note that  $R^{error}$  can contain estimates of both random (diagonal of matrix) and correlated errors (off diagonal), provided there is some quantified estimate or model of the correlated errors. In our case, we have no a priori knowledge of correlated errors so will assume random errors only, which is routinely done."

e) The resulting SSL, TSL, and OHC estimates are compared against an OI field based on Argo data only (the SIO estimate). Since Argo-only reconstructions are known to provide suboptimal representations of OHC and TSL this comparison is of limited relevance. Other products combining Argo and satellite altimetry for optimal interpolation exist (e.g., Lyman and Johnson 2008, the PMEL product; Lyman and Johnson 2023, the RFROM product) and they show significantly better results (Meyssignac et al., 2019 Frontiers; Hakuba et al., 2024 survey of geophysics). They should be included in the comparison.

**Response**: We cannot find the Lyman and Johnson, 2008 dataset on the PMEL website, nor can we find a link for it anywhere. We did find the MIMOC data (Schmidtko, S., G. C. Johnson, and J. M. Lyman (2013), MIMOC: A global monthly isopycnal upper-ocean climatology with mixed layers, J. Geophys. Res. Oceans, 118, 1658–1672, doi:10.1002/jgrc.20122), but our

understanding is this is a monthly climatology, so won't reflect the interannual variations as in the SIO data.

We did download the data from <a href="https://www.pmel.noaa.gov/rfrom/">https://www.pmel.noaa.gov/rfrom/</a>, but it is heat storage only and uses a eddy-covariance mapping, so any direct comparison with our maps will be primarily showing the differences between the eddy-scale signals and the longwave signals, so is not directly relevant. We have, however, included it in the comparison of global OHC and dOHC/dt in a new Figure (Fig. 11) and in subsequent discussion. See more in the specific comments below.

f) When making comparisons, the authors should separate the annual cycle from the interannual and longer-term variability (in particular in figures 5,6, 8 and 9). This is essential to assess whether the interannual and low-frequency components of SSL, TSL, and OHC are adequately resolved.

**Response**: We have added non-seasonal plots to all those figures (as well as included non-seasonal statistics in Table 1.

**2. The study lacks a validation section.**

The proposed estimates are not validated against any independent data. Validation is crucial to demonstrate the reliability of the results. A suitable validation approach would be a "leave-one-out" experiment, where the analysis is repeated multiple times excluding one or more Argo profiles; the omitted profiles can then serve as independent references. Another useful validation could involve comparing the global mean OHC estimates with surface flux-derived ocean heat uptake estimates from the combination of CERES TOA radiation (EBAF product) and the vertically integrated divergence of atmospheric energy (TDIV product), as in Mayer et al. (2022 JCLI). This method provides precise global-scale validation. To achieve a state-of-the-art assessment of OHC, both approaches should be implemented.

**Response**: Such a "leave-one-out" experiment would only measure short-wavelength variance differences between our longwave maps and the full profile SSL, TSL, etc. We would argue that the Altimeter-GRACE SSL global curves are essentially independent post 2010 when our mapping does not appreciably change under different reference grids (including with no reference) – see Figure 7 in the new version. This comparison is already included.

In response to other comments by the reviewer, we have eliminated a discussion of OHU, as dOHC/dt for the upper 2000m is only one (albeit, a major) component of it. Instead, we have focused only on OHC and dOHC/dt for the upper 2000m. In this case, we feel it is perfectly fair to compare with other Argo-based products to show where our results agree and disagree. We have, however, added another dataset, from the Lyman and Johnson (2023) analysis.

Detailed Comments:

Title and Abstract: The term "longwave length" must be clearly defined with a specific quantitative range.

**Response**: We have revised the abstract to define longwave (> 1500 km), which is based on a spectral analysis. We have added specifics in .

The study provides estimates of SSL, STL, and OHC only down to 2000 m depth. This critical limitation is not stated until line 404, which is too late and may cause confusion. It should be made explicit from the beginning.

**Response**: It has been added to the title and abstract.

Line 18: There appears to be a missing word. Please check the grammar.

Response: Yes, thank you. The missing word was observations.

Line 23: The term "deep ocean" may be incorrect in this context. You likely mean "open ocean."

**Response**: We meant ocean areas with deep water, but *open ocean* is better. Changed.

Line 29: Key recent literature is missing. Consider citing the following: Lyman and Johnson (2008 JCLI, 2023 JAOT), Meyssignac et al. (2019 Fronteirs in Marine science), Hakuba et al. (2024, survey of geophysics)

**Response**: We did not intend this list to be exhaustive, but to provide a list of papers over the past two decades. We have, however, added these additional studies.

Lines 28–42: The "non-exhaustive list" includes several datasets known to have biases due to missing state-of-the-art corrections (e.g., use of climatologies that relax to zero in data-sparse regions, as with Ishii et al.; or lack of salinity drift correction in Argo before 2019, as in EN4). At the same time, key datasets that use satellite altimetry for interpolation and are more accurate (e.g., Lyman and Johnson 2008; Hakuba et al. 2021; Marti et al. 2022 essd) are omitted. See Hakuba et al. (2024 survey of geophysics) for a comparative assessment.

**Response**: This list is for T/S gridded data from Argo. The Hakuba et al. (2021) does not provide a unique dataset (as far as we can tell from the data availability statement) and only uses (or combines other data) into integrated OHC, so it is not relevant. We cannot find the Lyman and Johnson (2008) dataset on the PMEL website, nor can we find a link for it anywhere. The Marti et al. (2022) data is an integrated global measure of OHU, not a gridded T/S product. So again, not relevant here.

We have added a new statement referring to maps of ocean heat content that are available: "Maps of ocean heat content, based on Argo and other data, are also routinely produced (Cheng et al., 2024; Lyman and Johnson, 2023)."

We have already noted that many of the products have deficiencies, but it is not within the scope of this paper to fully document those. We have included references to such studies for an interested reader. We have added Hakuba et al (2024) in the list noting the deficiencies for calculations of OHU.

Overall, we feel this is a fair description of other data products. We remind the reviewer that this manuscript is meant primarily to describe the methods for producing our dataset (the intent of ESSD and why we submitted the manuscript here) and not an exhaustive review of ALL Argobased data products and intercomparisons. As the review points out, there are already numerous papers describing these, which we reference.

Please also consider including: Cheng et al. (2024 essd) (IAP estimate)

Line 50: Marti et al. (2022 essd) also employ this technique. Please include this reference.

Line 58: A more recent paper by the same authors addresses deep and abyssal warming: Johnson & Purkey (2024 GRL)

Line 60: A Recent community effort compare various OHC estimation methods: Hakuba et al. (2024 survey in geophysics)

**Response**: References have all been added in appropriate places.

Lines 151–152: "Necessary corrections applied" is too vague. Specify whether corrections to altimetry are consistent with those in the GRACE mascon solution. In particular:

Was a consistent GIA correction used?

Were corrections such as Frederikse et al. (2017, GRL) applied to altimetry to address ocean crustal elastic response to contemporary ice mass loss?

**Response**: No GIA or current ice mass elastic responses have been applied to altimetry because such corrections are not available as maps to apply – they have be calculated and that is beyond our capability. Moreover, signals are tiny (< 0.3 mm yr-1) and so should not introduce significant errors. We have added a comment clarifying such corrections are not applied and why:

New text: "Note that while some authors (e.g., Frederiske et al., 2017) correct altimetry SSH for small changes in the ocean bottom from glacial isostatic adjustment (GIA) or present-day mass loss, this is non-standard and also not a large effect; values are of order < 0.3 mm yr-1. Within the period when altimetry will have a large effect on filling gaps in Argo data (2003-2007), overall changes will be less than 1 cm, which is smaller than the estimated uncertainty. Moreover, such corrections for altimetry are not provided as correction maps but have to be computed from an elastic earth model and ice loading histories, which is beyond the capability of the authors."

Line 202: Clarify what "normally" refers to. What standards or norms are implied? In which cases is temporal variability ignored, and how does that apply here? What are the implications?

**Response**: We think our comment is understandable indicating it means "all values within a certain period (e.g., within a calendar month for our calculations) are considered identically."

Lines 205–207: Using Wunsch (2003)'s method assumes uncorrelated signal and noise within the control radius (i.e.,  $\langle signal(i), error(j) \rangle = 0$  for all i, j). This assumption may not hold in practice. For example, salinity drift affecting profilers from the same manufacturer (deployed in the same regions) introduces a correlation between signal and error. This assumption must be explained and its limitations discussed in more details.

**Response**: As we noted in an earlier response, we have added text explaining why we do not use any correlated errors. If the reviewer knows of quantified covariance functions for errors in Argo floats, we would be happy to consider using them, but we do not know of any. In addition, because we use integrated SSL and TSL measurements, it would be difficult to convert a covariance for salt drift for a single float into the mapping OI function.

**Lines 216-217:**

Estimating the autocovariance from satellite data incorporates altimetry and GRACE error covariance into the interpolation error. In some regions, this can dominate the total error. Publicly available datasets provide uncertainty and error covariance for satellite data; They should be used to evaluate the impact. The authors should consider incorporating these error sources directly into the error budget (e.g., equations 8 and 9) for a more robust analysis. In the same way, the interpolation approach introduces the covariance of deep ocean signals—present in satellite data but missing in Argo—into the interpolation error in equation 11. This signal may be significant and temporally correlated (see Zilberman et al., 2024 GRL). This effect should be evaluated

And

Line 230: While random error may be assumed in equation 6, in reality, altimetry and GRACE data exhibit significant spatial and temporal correlations (see Prandi et al., 2021 Sci data). This correlation should be explicitly included in equation 8 via non-diagonal terms in C sat^error.

**Response**: Respectfully, we know of no public datasets that describe error covariances for either altimetry or GRACE maps in terms of either a covariance function in terms of radius, or a map. While plots have been presented for some corrections in altimetry (Prandi et al.), these maps are not available, only the plots in the paper! While mapped altimetry products are available, no error maps are provided! While there are some covariance matrices available for the TOTAL GRACE geoid in spherical harmonics, these do not represent the time-variable mass field. Nor are they in a useable form (i.e., for mascons).

Because such error covariances are not known, we stand by our calculations, as "incorrect" as they may be.

The same issue arises when sea level is used as a proxy for STL (line 290–291), especially in high latitudes where salinity signals are non-negligible. Please discuss.

Response: Yes, and this is why we stated, on lines 290-292: "This is a reasonable proxy in the midlatitudes (Chambers et al., 1997; Jayne et al., 2003), but less reasonable at higher latitudes because of larger salinity fluctuations there. This should be considered if using the thermosteric maps before more complete coverage of Argo in ~2008."

Line 233:The strategy of treating shortwave signals as correlated noise in order to smooth them out via the optimal interpolation scheme is central to the method. The author should give more details on how this works. They should expand on how exactly the approach work, which wavelengths are filtered and whether all shortwave signals are treated equally.

**Response**: This is merely a rearranging of the covariance functions and where they are evaluated. There is no change in total covariance structure modeled, so no need to discuss different wavelengths being "filtered" or treated differently. We have expanded our discussion of this for more clarity:

New text (**revision in bold**): "Random error is generally assumed in Equation (6), meaning the  $R^{error}$  matrix is diagonal only and the recovered maps will include the short-wavelength variance (albeit, with sampling error). However, there is nothing in formulation that requires that  $R^{error}$  only be random. In fact, some of the signal (for instance, the short-wave portion of the covariance) can be treated as a correlated "error" and so, the resulting OI map will

reflect the long-wave signal only. Note that in doing this, the  $\left[R_{jj}^{signal} + R_{jj}^{error}\right]^{-1}$  matrix in Equation (6) (i.e., the signal + error covariance evaluated for all the observation pairs in the window) does not change. If the  $R^{signal}$  matrix includes both the short and long covariances (so will have diagonal and off-diagonal terms), adding random error only adjusts the diagonal matrix. If, on the other hand, the short covariance structure is included in  $R^{error}$ , then when added together, one gets the same matrix. The difference arises in the  $R_{ij}^{signal}$  matrix (i.e., the matrix containing the covariances based on the distance from the observations to the center point of the grid). When short (eddy) covariance structure is included in  $R_{jj}^{error}$ ,  $R_{ij}^{signal}$  will only contain longwave covariance structure. Thus, when  $\alpha$  is calculated, it will contain the weights to map only the longwave portion, while the shortwave and random parts of the covariance are accounted for in the error.

In our case, then:

$$C_{signal}^{sat}(r) = 25e^{-\left(\frac{r}{1675\,km}\right)}$$

$$C_{error}^{sat}(r) = 82e^{-\left(\frac{r}{100 \text{ km}}\right)^2} + 25 \text{ cm}^2 \text{ (random)}.$$
 (8)

and so  $R_{ij}^{signal}$  and  $R_{jj}^{signal}$  will be computed from  $C_{signal}^{sat}(r)$ , while  $R_{jj}^{error}$  is calculated from  $C_{error}^{sat}(r)$ . However, the overall covariance function remains the same; it is merely partitioned differently between "signal" and "error."

*Line 260: You are likely referring to equation 6 here, not equation 5.*

**Response**: Yes. Thank you.

Line 261: Since Argo profiles OI is used here, the error covariance should reflect the structure of Argo data signal and error, not satellite data signal and error. Please address this inconsistency or discuss the limitations implied by replacing one by the other

**Response**: As we stated, the covariance structure in Equation (11) reflects the "the autocovariance structure of Argo profile SSL after removing the long-wave mapped satellite SSL." We did test this. I actuality, the roll-off parameters changed slightly, but not enough to warrant different values in our opinion. We have added some text to clarify this:

New text: "Note that in Eq. (11), the shapes of the autocorrelation functions are identical to those used in the satellite mapping in terms (accounting for the short- and long-wavelength component). Only the variance at lag-0 has changed, to reflect the reduction in variance by first removing an a priori reference value, leaving only residuals to map. While the actual "best-fit" roll-off parameters estimated to the residuals is slightly difference from those estimated from the altimetry, they were close to those estimated for the satellite data (< 10 km for the eddy-scales and < 200 km for the long-wave). Testing with the actual roll-off values versus the consistent roll-off values led to differences far less than estimated errors (or order 5 mm), so we chose to use consistent roll-off values."

Line 282: Step 3 removes any deep ocean signal present in altimetry/GRACE that is absent in Argo (limited to 2000 m). This makes it clear—albeit belatedly—that SSL, STL, and OHC are limited to 0–2000 m. This should be stated clearly and early in the manuscript. The implications are significant: the dataset cannot capture full-depth ocean heat uptake. This other limitation must be acknowledged and discussed.

Response: Step 3 is not intended to remove the deep signal (which is only a small bias), but because different time-references were applied to create anomalies for altimetry, GRACE, and Argo. This creates much larger differences in the upper 2000 m (e.g., averaging over different ENSO phases in the Pacific). As we clearly state in the paragraph before: "If one does not reference the  $\Delta SSL_{sat}$  reference grids to a comparable time-period as Argo, this would potentially cause biases from 2003 to about 2008 when Argo data is not complete (Fig. 1, Fig. S1)"

As noted previously, we have added multiple statements (including the abstract and the title).

Line 285: The equations become unclear here. For instance,  $\Delta SSL\_sat^oi\_corr(x,y,t)$  is referenced but never appear in the equations. Likely, it should appear in equation 10, but this is not explicit. A clear workflow diagram with consistent notation would certainly adress this issue. And by clarifying the actual overall computation process, it would greatly improve reproducibility.

**Response**: It DOES appear in Equation (10), just without the (x,y,t) for simplicity, which we felt would be obvious to any reader. We also switched from longitude, latitude, time, to x,y,t. For complete clarity, we have revised the equations to explicitly put (lat, lon, t) for every parameter in eq. (10) and throughout the list of steps for correcting for the bias.

Line 305: In the figure 2 caption, you likely meant "thermosteric sea level" instead of "steric sea level"

**Response:** Actually, this is used to convert satellite SSL maps to OHC. As we note, this is not the perfect (as SSL can differ from TSL), but it is the best approach. We have changed the caption to read "satellite SSL grids".

Line 329: High correlation values mostly reflect the agreement between annual cycles. Readers also need to know what the correlation numbers are when the annual and semi-annual signal are removed. This also applies to figures 5, 6, 8, and 9.

**Response:** We have added statistics when the annual signals (and trend) are removed to Table 1 and have added panels where the annual variations are removed to Figures 5, 6, 8, and 9. Note in doing this, we decided to also redefine the basins to not include the tropics separately, and now define what we mean by North and South basins. Results are essentially the same, with actual reduction in the residual standard deviation primarily from removing rend errors. We also include some notes in the revised text on calculations made over a smaller area.

Line 331: The statement that "longwave maps are sufficiently accurate to resolve global and regional sea level budgets" is overstated. Accuracy depends on the specific scientific application. If the goal is to identify contributions to sea level rise, these maps may not be suitable, as they cannot estimate deep ocean warming or confirm its absence. Please revise this claim accordingly.

**Response:** Edited to now read "Thus, we conclude the longwave mapping is sufficiently accurate to resolve average monthly SSL, TSL, OHC anomalies for the upper 2000m over areas greater than 1500 x 1500 km. At smaller areas, especially within eddy regions, the longwave maps cannot fully resolve the average and should not be used for studies over regions smaller than this."

Line 335: The caption of figure 3 is unclear. Clarify what is meant by "eddy covariance as signal" vs. "eddy covariance as error (longwave signal)."

**Response:** We have changed the caption to reference appropriate equations. New caption read: "Standard deviation of Argo sampling experiments, where altimetry interpolated to Argo profile locations and times is mapped and compared to maps from the full altimetry. a) Eddy covariance as modeled as signal (Eq. (7)) and b) eddy covariance as modeled error (Eq. (8)), i.e., the longwave mapping."

Line 395: If only one dataset is used for comparison, the results should ideally be compared with Lyman and Johnson (2008 JCLI) or Lyman and Johnson (2023 JAOT), which also used altimetry to interpolate Argo profiles. This would assess whether the current estimate matches the state of the art and whether adding GRACE data improves accuracy.

**Response:** The Lyman and Johnson (2023) data are OHC anomalies, not SSL or TSL, so we would have to convert under certain assumptions to convert. We have included a comparison with that data to in a new global OHC and dOHC/dt plot (new Figure 11), which is fairer and more consistent. Note that here, the Altimetry-GRACE estimate is probably independent post-2010, when the USF maps (at least) appear to be reference independent because of the number of Argo profiles (e.g., Figure 6 (Figure 7 in revision)).

Line 414 & Figure 9: TSL from USF and SIO aligns better with altimetry+GRACE-derived SSL than their own SSL estimate. This implies persistent salinity issues, even in recent data. Please comment.

**Response:** We did comment on that we wrote it was "due to unresolved salinity errors that are not flagged (and so get into the USF processing) but that SIO has edited out." We have expanded this in the revision.

New text: "Moreover, when global thermosteric sea level (GTSL) is compared instead (**Fig. 10**), there is no change in behavior during 2015, but USF and SIO GTSL agree well for all other periods. This suggests that the difference between USF and SIO GSSL in 2018-2020.5 is due to unresolved salinity errors that are not flagged (and so get into the USF processing) but that SIO has edited out. Unfortunately, there is no documentation that we can find on recent SIO processing standard that can explain this. However, the differences in 2015 can't be explained by salinity errors, as they appear in both the USF GSSL GTSL curves."

Line 429: Note that actually an open international comparison effort is ongoing: ME4OH, led by M. Palmer (UK Met Office) and D. Giglio (NCAR) in which all raw data from different groups is available.

**Response:** The MapEval4OceanHeat initiative is using synthetic T/S profiles derived from a model for an intercomparison of mapping methods. While that is a great initiative (for all groups to test mapping from the same original data), our point here is that we don't even know which Argo profiles are used by all the groups. Did some use all available Argo floats without considering the flags? Are some floats edited out because they have an algorithm to detect the salinity sensor problems? Are adjustments made? This is the dataset that would be useful to

share, and the point we were making. Because the MapEval4OceanHeat initiative is doing something else, we do not believe it is relevant to mention here.

Line 433: OHU cannot be estimated accurately here, as the analysis omits OHC below 2000 m. This must be stated explicitly.

**Response:** We have stated this explicitly now, and comment on how the derivative of OHC only captures part of OHU.

New text: "A common use of OHC anomalies for the upper 2000m is to take the time-derivative of the global average (in W m-2) and use this in computing the global ocean heat uptake (OHU) (e.g., Hakuba et al., 2021, 2024; Loeb et al., 2021; Lyman and Johnson, 2023). The upper ocean above 2000m explains approximately 90% of the OHU, while the ocean deeper than 2000m explains the remainder (Purkey and Johnson, 2010; Hakuba et al., 2021). Importantly, interannual to decadal-scale changes in OHU are dominated by the changes above 2000m. Typically upper ocean OHC derivatives are combined with an estimate of deep warming below 2000m (assuming a steady gain of heat), using values determined from deep and repeat hydrographic sections (e.g., Purkey and Johnson, 2010; Johnson and Purkey, 2023). The value commonly used added is  $0.06 \pm 0.04$  W m-2 (e.g., Hakuba et al., 2024). Additionally, the OHC derivatives are often normalized by Earth's surface area at the top of the atmosphere (5.14 × 1014 m2 at 20 km above the Earth's surface) and not the ocean area, to bring it into line with satellite measurements of Earth's Energy Imbalance (EEI) (e.g., Hakuba et al., 2021; Loeb et al., 2021)."

Furthermore, we no longer use OHU in Figure 10 (now Figure 11). Instead, we refer to it as dOHC/dt for the upper 2000m. We have also added the series from Lyman and Johnson (2023) in the plot (and also show mean OHC in ZJ). This led to some additional discussion:

New text: "Here, we compare the global OHC anomalies above 2000m computed from our OHC grids with those from computed from the SIO T/S grids as well as those computed by Lyman and Johnson (2023), which use altimetry and satellite sea surface temperature measurements as a reference before combining with Argo (**Fig. 11a**). Before 2015, the three datasets agree well, showing similar overall trends and interannual variability. While the Lyman and Johnson (2023) grids are specifically formulated to resolve eddy signals, it is clear this has little effect on the global average and that the longwave mapping we use is sufficient. within estimated errors, except for the periods that include 2015 (**Figure 11**). During 2015 to early 2016, there is disagreement as noted previously (e.g., **Figs. 9 & 10** and associated discussion). The USF OHC appears to be the outlier in 2015, but by early 2016, it agrees with the PMEL OHC, whereas the SIO OHC has a significant drop throughout 2016 until 2017. This further supports the idea that there may be unresolved issues with Argo temperatures in some floats in 2015 and 2016 that warrants further investigation.

After 2020, there is a significant change in behavior between the USF/SIO OHC curves and that from the Lyman and Johnson (2023) analysis. The PMEL analysis shows a steady rise in OHC after 2020, whereas USF and SIO grids indicate more interannual variability, with a drop from 2020-2021, followed by a subsequent rise. It is interesting that the PMEL curve follows the general trend in satellite altimetry over this time (e.g., **Fig. 10**), which suggests that the global

OHC from the PMEL analysis may be more dependent on the altimetry reference than our analysis, as we find little to no impact of different references in the global average (**Fig. 7**). Notably, the USF curve post-2020 follows that of SIO, which uses only Argo data (and an Argobased climatology) in the mapping."

Figure 10: The figure would be more convincing if compared with independent OHU estimates such as Mayer et al. (2022 JCLI), which use CERES and ERA5 data.

**Response**: As the reviewer as noted several times, our estimates only show dOHC/dt above 2000m and NOT the entire OHU. We have removed any methion of OHU from the manuscript, other than to provide a general idea that dOHC/dt above 2000m is an important part of it.

Because of this, and because the Mayer et al. or CERES-only fluxes will provide the total OHU, so would have to have estimations of deep warming removed to be comparable, we do not believe such a comparison is warranted. We have, instead, added another independent Argosatellite based estimate of OHC above 2000m from Lyman and Johnson (2023) to the analysis.

Line 462: I think the statement "useful benefits for scientists working on sea level-budget and ocean heat uptake studies" is overstated and not supported by the analysis. The deep ocean contribution to sea level is arguably the most important unknown contribution (in amplitude) to the sea level budget. As deep ocean estimate is simply lacking here, the benefit for sea level budget science is limited. Even regionally the absence of deep ocean estimate is very limiting. Recent studies based on deep argo reveal that deep ocean signal is actually a significant contributor to the regional sea level variability (up to 30% in some regions, Zilbermann et al. 2024 GRL Johnson et al. 2019 GRL). As for the OHU studies, this is impossible to do with the proposed dataset as it does not provide any estimate of the OHU it only provides an estimate of the ocean heat content changes from 0 to 2000 m depth.

**Response**: The same could be said for any current Argo-based product, but they are still used routinely for global and basin-scale sea level budget studies. We have added a comment on this at the end, acknowledging the very still open research topic of deep steric signals, but also pointing out our maps are as useful as any current Argo-based maps being used for these types of studies.

New text: "There are limitations to the data. These maps are intended to represent SSL, TSL, and OHC for the upper 2000m of the ocean. They do not account for any steric variations below 2000m. However, this is true of any other current Argo-based mapped product of T/S or OHC above 2000m. New data from the deep Argo array indicate the possibility of relatively large steric anomalies below 2000m in at least one small area of the tropical Atlantic (Zilberman et al., 2025), but complete knowledge of deep steric signals is still very much an open research area. The maps we have produced will be as useful as any other current Argo-based product for global and basin-scale sea level and heat budgets that are available at the present, provided the studies are for areas larger than about 1000km by 1000km."

---

## Author Comment (AC2)

We would like to thank **Reviewer # 2** for their thorough review of our manuscript and thoughtful suggestions. Below, we respond to the more significant suggestions, in the order they appear. The reviewer's comment is in italics, while our **Response** follows.

This study proposes a new data set of steric sea level and ocean heat content (OHC) based on Argo data for the 0-2000m depth range and 2003-2023 time span. From my understanding, the two main differences with other existing Argo-based steric data sets are: (1) the use of a 'geodetic' steric climatology as reference-or first guess- (defined as the difference between the altimetry-based global mean sea level/GMSL and GRACE-based ocean mass), and (2) a long-wavelength mapping for reducing uncertainty due to meso-scale eddies not well observed by Argo.

Such a new data set could be of great value for some studies that investigate the long-term upper ocean warming. It could be also considered as a useful new indicator of the contribution of the upper ocean to the Earth energy imbalance. However (and this is not mentioned in the manuscript), I would be more cautious about using it for assessing global and regional sea level budget closure, as explained below.

While such a data set may be worth to be published, the manuscript needs substantial revision and clarification.

• The authors do not explain well the novelty of their results neither the specific applications of their data set. To me, such a data set would be useful as an indicator of upper ocean warming but should not be used for sea level budget closure assessments. In effect, the use altimetry-based GMSL corrected for the ocean mass contribution (equivalent to the steric sea level) as first guess (or reference) will have a strong weight during the first 5 years of the record in any sea level budget assessment because of the poor Argo data coverage, hence the limited information brought by Argo data in the optimal interpolation process. Over this time span, the computed steric sea level will thus be dominated by the 'Altimetry minus GRACE' signal. As a result, one may expect that the GMSL budget of the early part of the record will be artificially closed because the estimated steric sea level from this study will be - by construction- essentially equal to the 'Altimetry minus GRACE' time series.

**Response:** This dataset will be as useful in sea level budget studies as any other Argo-based T/S gridded product. Such Argo T/S products for the upper 2000m are routinely used in sea level budget analyses. While our grids will default back to the altimetry-GRACE reference in early years (strongest before 2005, reducing from there to about 2008 as demonstrated in the analysis), the Argo data controls the mapping post-2008 (as we discuss).

We have revised the manuscript to make it clearer that these products are intended to provide SSL, TSL, and OHC anomalies above 2000m. For example, in the discussion section, we now state:

"There are limitations to the data. These maps are intended to represent SSL, TSL, and OHC for the upper 2000m of the ocean. They do not account for any steric variations below 2000m.

However, this is true of any Argo-based mapped product of T/S or OHC above 2000m. New data from the deep Argo array indicate the possibility of relatively large steric anomalies below 2000m in at least one small area of the tropical Atlantic (Zilberman et al., 2025), but complete knowledge of deep steric signals is still very much an open research area. The maps we have produced will be as useful as any other current Argo-based product for global and basin-scale sea level and heat budgets that are available at the present, provided the studies are for areas larger than about 1000km by 1000km."

• The term 'long-wavelength' is unexplained (500 km?, 1000km? more?...).

**Response:** Apologies. We assumed this would be clear from the equations used for the OI. We have added text after Equation (7) to be explicit about what we mean by "longwave" and "shortwave."

"Throughout the remainder of this paper, when we say "longwave" or "long wavelength" we mean the portion of SSL that has a covariance function equal to  $25e^{-\left(\frac{r}{1675\,\mathrm{km}}\right)}$ , i.e., an exponential decay with a roll-off of 1675 km and max covariance of 25 cm². By "shortwave" or "short wavelength" we mean the portion of SSL that has a covariance function equal to  $82e^{-\left(\frac{r}{100\,\mathrm{km}}\right)^2}$ , i.e., a Gaussian with a roll-off of 100 km and max covariance of 82 cm². By "eddy resolution" we mean the full covariance structure of Equation (7). A spectral analysis of the longwave maps compared to the original, unsmoothed data indicates the longwave maps keep nearly 100% of the power for wavelengths longer than 1500 km."

We have also added another line to equation (7) to make this separation abundantly clear in the equation:

$$C^{sat}(r) = 82e^{-\left(\frac{r}{100 \, km}\right)^{2}} + 25e^{-\left(\frac{r}{1675 \, km}\right)} , \qquad (7)$$

$$C^{sat}(r) = C^{sat}_{short}(r) + C^{sat}_{long}(r).$$

We also now state the definition of long-wavelength in the abstract, and verify that averages over a box that is 1500km by 1500km in an eddy-rich region is consistent with the average over the raw data in the same region:

New test on lines 439-445 of the revised text: "We performed an additional test over a smaller area (1500 km by 1500 km) east of Japan where there is significant eddy activity. Correlations remain 0.99 or higher (even after removing a trend and annual variations) and standard deviation of the differences only increases to 2 mm, with trend errors < 0.2 mm yr-1. However, if smaller areas are considered (for example 500 km by 500 km in the same area), non-seasonal correlation drops to 0.8 and the standard deviation of differences increases to 4 cm, with significant trend differences (1.7 mm yr-1). Thus, we conclude the longwave mapping is sufficiently accurate to resolve average monthly SSL, TSL, OHC anomalies for the upper 2000m over areas greater than 1500 x 1500 km. At smaller areas, especially within eddy regions, the longwave maps cannot fully resolve the average and

should not be used for studies over regions smaller than this."

• Above all, the results need thorough validation in order to convince users about the interest of using this new data set. Only a single comparison with one Argo product is proposed in the manuscript. More is needed. For example, the derived OHC data set could be compared with CERES data.

**Response**: We would argue that the Altimeter-GRACE SSL global curves are essentially independent post 2008 when our mapping does not appreciably change under different reference grids (including with no reference) – see Figure 7 in the new version (Figure 6 in old version). This comparison is already included in the analysis.

In response to comments by **Reviewer # 1**, we have eliminated a discussion of OHU, as dOHC/dt for the upper 2000m is only one (albeit, a major) component of it. Instead, we have focused only on OHC and dOHC/dt for the upper 2000m. In this case, we feel it is perfectly fair to compare with other Argo-based products to show where our results agree and disagree. We have, however, added another dataset, from the Lyman and Johnson (2023) analysis.

Because the CERES-based estimates of OHU (such as those calculated by Mayer et al) would include portions our maps do not measure (such as OHU below 2000m or in the Artic Ocean). While we could add estimates from other studies (which is done in any Argo-based estimate of OHU), we feel showing the estimates of OHC and dOHC/dt for the upper 2000m against two other Argo-based products is sufficient.

• It is a pity that all presented steric sea level curves include the seasonal cycle while one is likely more interested in the interannual variability. The discussion about trend differences is quite superficial.

**Response**: The annual/semiannual signals have been removed and interannual variations (unsmoothed) are shown for all time-series now as a separate panel. We believe it is also important to show that the seasonal variation is captured as well, which is why it remains.

Statistics for the basin-averages in Table 1 are now computed for the full series and the interannual-only ones. Note that removing the seasonal variation has no impact on correlation and the standard deviation of residuals becomes smaller, reflecting that much of the difference arises from small changes in the seasonal variation, not the interannual variations.

Additional analysis on the interannual variations has been added throughout the revised manuscript.

• Finally, while the authors are well aware of the Argo-based salinity measurement errors as of 2015, there is no discussion on the reliability of their halosteric data set. This is an important limitation of this study.

**Response**: We stated that there are clearly residual errors in the SSL data that are most likely from salinity, even though we use the latest version of the Argo profile data with new flagging and adjustments (e.g., Wong et al., 2023). We have revised this section to make this clearer. In addition, in the Methods section, we have explicitly stated that the flagging we use should eliminate floats with detected salinity drift using the methods of Wong et al. (2023).

New text: (in Section 2.1, **new text is bold**): "Additionally, profiles were retained only if the QC flags for position, date, pressure, salinity, and temperature were all set to "1," indicating the highest quality measurements (Wong et al., 2023). **In particular, this flag indicates no sign of apparent salinity drift in the float.**"

New text: lines 595-603 in revision: "Moreover, when global thermosteric sea level (GTSL) is compared instead (**Fig. 10**), there is no change in behavior during 2015, but USF and SIO GTSL agree well for all other periods. This suggests that the difference between USF and SIO GSSL in 2018-2020.5 is due to unresolved salinity errors that are not flagged (and so get into the USF processing) but that SIO has edited out. Unfortunately, there is no documentation that we can find on recent SIO processing standards that can explain how this was accomplished. Moreover, the SIO and USF GSSL curves both depart significantly from the altimetry-GRACE curve after 2020, whereas the SIO and USF TSL curves differ by a smaller amount. This further suggests unresolved salinity errors in a large number of profiles after 2018, even though we have utilized a release of Argo T/S profiles that has added new flags and adjustments to identify such problem floats (Wong et al., 2023). We removed such flagged/adjusted floats from our processing stream (see Section 2.1). However, this does not prevent large and unrealistic global halosteric signals post 2018, so we must conclude that unidentified salinity errors remain in enough profiles to affect the global halosteric signal."

**Detailed comments**

• Abstract, line 9: indicate the wavelength range. What means 'long-wavelength'?

**Response: Done.**

• Abstract, line 11: explain what is a 'mean climatology'

**Response**: Changed to read: "...the satellite estimate is used instead of a mean monthly climatology or zero, as other analysis centers use."

• *Introduction, line 23: I suppose that you mean 'open ocean' rather than 'deep ocean'?*

**Response: Yes. Changed.**

• Introduction, lines 49-62: indicate here that your altimetry data set does not account for the Jason-3 radiometer drift (a problem that may impact your results, especially at the end of the record)

**Response**: We feel this is more relevant in the data section than the Introduction, so have added it there. However, we note that not using the radiometer drift correction will not affect the final grids, as this only affects data after 2016 and that in this period the mapping is controlled by Argo, not altimetry-GRACE (see Figure 7). Also, we do use the radiometer drift correction for Figures 9, 10, and 11 (and we already note that in the text).

New text in Section 2 reads: "We note that these data do not include a correction for the recently discovered small drift of the Jason-3 radiometer (Brown et al., 2023), which will affect the altimeter after January 2016. However, during this time, Argo sampling is of sufficient density (Fig. S1 and S2) that they should correct for any small drift errors in the reference surface used. We will demonstrate this is so in Section 3 by utilizing different initial reference grids, including zero and a mean monthly climatology, neither of which are affected by any radiometer drift."

We considered re-doing our analysis when the newest Beckley et al analysis was released which included the J3 radiometer drift correction. But a quick test with a couple of months indicated no change in the final grids.

• Introduction, lines 57-58: deep ocean warming may no more be negligible (e.g., Johnson and Purkey, 2024).

**Response**: We have deleted that sentence and rephrased it to better emphasize this is still very much an open research question and that groups have to add in an estimate of SSL contributions below 2000m to study the sea level budget.

New text: "Fully quantifying deep ocean contributions to steric sea level is still very much an open area of research (e.g., Purkey and Johnson, 2010; Desbruyères et al., 2016; Johnson and Purkey, 2024). For example, a recent study suggests that short-term changes in small areas may be of the order of ± 2 cm for periods of up to a year (Zilberman et al., 2025), Because of this, studies that study the sea level budget have to add an estimate of deep steric changes to any Argo-based product that can only resolve changes above 2000m. With that caveat, numerous studies have compared the two methods on regional and global scales, finding good agreement in ocean heat content (von Schuckmann et al., 2014; Hakuba et al., 2021; 2024; Marti et al., 2022; Meyssignac et al., 2019; ) and global thermosteric sea level (Blaquez et al., 2018; Barnoud et al., 2021), but with substantial differences in steric sea level after 2015 (Blaquez et al., 2018; Chen et al., 2020; Barnoud et al., 2021), likely due to salinity problems in the gridded Argo T/S products."

Again, this is not a unique problem to the data we have created, but is true of ANY Argobased SSL estimate.

• Section 2.1, lines 116-118: the sentence 'we kept only profiles deeper than 750 dba' is unclear. Which depth? Why?

**Response**: 750 dbar is the pressure level (so  $\sim$ 750 m depth). This is actually a typo – we used an initial selection to 750 dbar, but further restricted the max to 1000 dbar in the gridding step. We have revised the sentence to be clearer on this, and added a note on a test where we found it made little difference on the mapping using some floats to 1000m and some to 2000m, or all to 2000m (really, anything deeper than 1000m).

"Since our goal is to integrate vertically to obtain a value for upper ocean  $\triangle SSL$ , etc, we kept only profiles that had a maximum pressure/depth greater than 1000 dbar, had a minimum pressure  $\leq$  50dbar, and that had more than 50 observations in each profile. Although we allowed profiles that only sampled the upper 1000m and not the full 2000m of the upper ocean, this was a relatively small number and tests indicated it did not significantly alter recovered maps over using a restriction to profiles that observed down to > 1900 dbar."

• Section 2, line 145, I suppose that 'who which' should read 'who wish'...

Response: Yes. Changed.

• Section 2.2: line 152: indicate which corrections are applied (and mention that the Jason-3 radiometer drift is not accounted for)

**Response**: We note first that we don't make corrections. These are done by Beckley et al. and their final product is fully corrected SSH anomalies. It is beyond the scope to provide a full description of their processing (or specific models used), but we have added a note about some of the main ones and referred the reader to the appropriate document for more info. We have also added the note about the radiometer.

New text: "These data have consistent geophysical corrections and orbits and have had all standard geophysical corrections applied (e.g., inverted barometer, ocean tides, wet and dry troposphere, ionosphere, sea state bias) as described in the data documentation (Beckley et al., 2010; 2022). We note that these data do not include a correction for the recently discovered small drift of the Jason-3 radiometer (Brown et al., 2023), which will affect the altimeter after January 2016. However, during this time, Argo sampling is of sufficient density (Fig. S1 and S2) that they should correct for any small drift errors in the reference surface used. We will demonstrate this is so in Section 3 by utilizing different initial reference grids, including zero and a mean monthly climatology, neither of which are affected by any radiometer drift."

• Section 2.2: lines 173-174: mention (for readers not familiar with the sea level budget) that 'altimetry-based sea level minus GRACE-based ocean mass' represents the steric sea level

**Response**: We stated this multiple times throughout the Introduction and do not feel it is necessary to repeat it again here in the data section. We feel it is well established in our manuscript that we are using altimetry-GRACE for the satellite estimate of SSL.

• Section 2.3, Optimal interpolation: this section is not easy to read. The numerical values mentioned in the equations seem to have been pulled out of a hat...

**Response**: Numbers are not "pulled out of a hat." We explained the process to compute the rolloff and amplitudes for the Gaussan and exponential functions of the covariance right before Equation (7):

"A single autocovariance was computed for all months between 2003 and 2023 – we tested computing month-specific functions, but the differences were minimal, so we used the single covariance function to reduce complexity. We then approximated the covariance values with a continuous function comprised of a Gaussian for short wavelength signals and exponential decay functions for the longwave portion, along with a random component to match the full variance of the data, similar to that done in previous studies (e.g., Willis et al., 2008; Roemmich and Gilson, 2009). Optimal parameters for the roll-off and amplitude parameters were estimated using non-linear least squares based on iterating values of the roll-off parameters over a range of expected values. "

• Section 2.3, line 215: what means '4-10x'?

**Response**: Sorry. "x" in this context is often used for "times" and that is what we meant: 4-10 times slower. We have changed it to "times."

• Section 2.3, line 272: what means 'mean surface' for GRACE data? (and Argo); line 289: 'corrected' for what?

**Response**: All 3 datasets (GRACE, altimetry, Argo) are anomalies relative to the mean state for some period. If that state were fixed and not evolving in time, this would not be a problem. But, for example, the mean ocean surface in 2000 is (on average) than the mean in 2010, or 2012. Averaging of climate patterns like El Niño can also cause regional biases differences. We have revised this section to emphasize these data are all anomalies computed relative to different mean periods.

New text: "However, before combining the satellite and Argo data in the OI scheme, one must account for differences in the reference period used for the satellite altimetry, GRACE/FO, and Argo anomalies. For example, GRACE/FO anomalies are referenced to a 2005-2010 mean period, the altimetry SSH anomalies to a mean period of approximately 1993-2018, and Argo to T/S means for 2004-2018."

• Section 3, Table 1: since you do not remove the seasonal cycle, the correlation essentially refers to this dominant seasonal cycle. What about the correlations at interannual time scale?

**Response**: Correlation of interannual (seasonal removed) and standard deviation of interannual residuals has been added to Table 1. In all cases, there is no significant difference in interannual versus full comparisons.

• Section 3, Table 1: trend differences around 0.3 mm/yr are not negligible at all (same order of magnitude as the GIA signal in the altimetry-based GMSL)

**Response**: These were typos. In recomputing the data for the new Figure 1, we discovered we had transcribed differences in the bias term, not the trend term, from the calculation to the Table. These have all been corrected. New maximum values are  $\pm 0.18$  mm yr-1, which are well within the uncertainty of the GMSL trends from satellite altimetry.

• Section 3, Figures 5 and 6: Show difference time series!!! Remove the seasonal cycle

**Response**: The SSL and TSL figures now also include a panel showing the interannual variations after removing the best seasonal fit. We choose not so show differences as well, as these can be deduced from the plots we show.

• Section 3, lines 59-362: The trend difference over 2005-2024 (1.04 versus 1.17 mm/yr) is not significantly larger than the one over 2011-2024 (0.08 mm/yr), so that the use of the geodetic reference has limited impact.

**Response**: Possibly, but it does have an impact and the difference is outside the uncertainty, so does show the impact on GMSL of the switch between a climatology and the satellite reference.

• Section 3, results: Why not show maps of regional trend differences? I suppose that the regional trend differences are not uniform but may depend on the regional Argo data sampling

Response: Comparing regional trend differences between our grids and those from, for example, SIO, would mainly emphasize mesoscale eddies, so are not relevant. And comparisons with altimetry-GRACE would show general differences between the satellite and Argo SSL products, which has been described by several other groups. This is also more of a scientific investigation, rather than a data description paper, which is the purpose of this manuscript, and why it was submitted to Earth System Science Data. We have, however, added a plot of standard deviation of differences between the altimetry-GRACE reference and final grids (as well as with an altimetry reference) (new Figure 3) at the request of Reviewer #1, which provides some information on the agreement.

• Section 3, line 380: 'The satellite data tend to dominate the mapping from 2003 to 2008...': see my comment above

**Response**: This sentence was discussing where the monthly maps are based on the satellite reference or Argo data, not the impact of the reference on the calculation of the global average, and will affect areas with significant interannual variability. We have added a sentence commenting on this:

New text: "This should result in an improved estimate of SSL, TSL, and OHC in these areas over using a mean monthly climatology if there are significant interannual fluctuations

there – e.g., El Niño/La Niño signals in the eastern Pacific, which did not have complete Argo coverage in every month until 2008 (Fig. S1)."

• Section 3, lines 400-404: the discussion about the corrections should appear much earlier in the manuscript (see comment above)

**Response**: We have added the note on the radiometer correction earlier and why it was not applied in the original processing but why it is here.

• Section 3, figures 8 and 9: show difference time series!!! Remove the seasonal cycle

**Response**: The figures now also include a panel showing the interannual variations after removing the best seasonal fit. We choose not so show differences as well, as these can be deduced from the plots we show.

• Section 3 figures 8 and 9: what is the impact of salinity measurement errors on the USF SSL curves? Please discuss

**Response**: we felt we did discuss this, but have added some more discussion in the revision:

New text: "This suggests that the difference between USF and SIO GSSL in 2018-2020.5 is due to unresolved salinity errors that are not flagged (and so get into the USF processing) but that SIO has edited out. Unfortunately, there is no documentation that we can find on recent SIO processing standards that can explain how this was accomplished. Moreover, the SIO and USF GSSL curves both depart significantly from the altimetry-GRACE curve after 2020, whereas the SIO and USF TSL curves differ by a smaller amount. This further suggests unresolved salinity errors in a large number of profiles after 2018, even though we have utilized a release of Argo T/S profiles that has added new flags and adjustments to identify such problem floats (Wong et al., 2023). We removed such flagged/adjusted floats from our processing stream (see Section 2.1). However, this does not prevent large and unrealistic global halosteric signals post 2018, so we must conclude that unidentified salinity errors remain in enough profiles to affect the global halosteric signal."

• Section 3, figure 10: there is no explanation for the 2015 difference between USF and SIO OHC. Please discuss

**Response**: We did discuss the 2015 difference in OHC, tying it to the observed difference in TSL (since TSL is proportional to OHC). We have added a new OHC and dOHC/dt curve from the PMEL analysis and have added even more discussion than in the original draft:

New text: "Here, we compare the global OHC anomalies above 2000m computed from our OHC grids with those from computed from the SIO T/S grids as well as those computed by Lyman and Johnson (2023), which use altimetry and satellite sea surface temperature measurements as a reference before combining with Argo (**Fig. 11a**). Before 2015, the three datasets agree well, showing similar overall trends and interannual variability. While the

Lyman and Johnson (2023) grids are specifically formulated to resolve eddy signals, it is clear this has little effect on the global average and that the longwave mapping we use is sufficient. within estimated errors, except for the periods that include 2015 (**Figure 11**). During 2015 to early 2016, there is disagreement as noted previously (e.g., **Figs. 9 & 10** and associated discussion). The USF OHC appears to be the outlier in 2015, but by early 2016, it agrees with the PMEL OHC, whereas the SIO OHC has a significant drop throughout 2016 until 2017. This further supports the idea that there may be unresolved issues with Argo temperatures in some floats in 2015 and 2016 that warrants further investigation.

After 2020, there is a significant change in behavior between the USF/SIO OHC curves and that from the Lyman and Johnson (2023) analysis. The PMEL analysis shows a steady rise in OHC after 2020, whereas USF and SIO grids indicate more interannual variability, with a drop from 2020-2021, followed by a subsequent rise. It is interesting that the PMEL curve follows the general trend in satellite altimetry over this time (e.g., **Fig. 10**), which suggests that the global OHC from the PMEL analysis may be more dependent on the altimetry reference than our analysis, as we find little to no impact of different references in the global average (**Fig. 7**). Notably, the USF curve post-2020 follows that of SIO, which uses only Argo data (and an Argo-based climatology) in the mapping.

These subtle differences in OHC are reflected in the time-derivative (**Fig. 11b**), which we calculated using running 2-year trends (along with annual and semiannual sinusoids) from the global average OHC (and converting J m-2 yr-1 to W m-2) – the time stamp used is the middle of each 2-year window and a one-month step was used. Values agree reasonably well before 2020 (noting the small differences in 2015-2017 noted earlier). The mean values for 2005 to 2023 are similar for USF and SIO (USF:  $0.58 \pm 0.18$  W m-2; SIO:  $0.54 \pm 0.21$  W m-2) but are significantly higher for the PMEL OHC derivative ( $0.96 \pm 0.19$  W m-2). This is primarily caused by higher values post 2020. In the first half or of the record (2005-2015), the mean PMEL values of dOHC/dt are 0.63 W  $\pm 0.24$  W m-2, whereas in the second half (2015-2024), the mean of the PMEL series is  $1.36 \pm 0.20$  W m-2."

• Section3, figure 10: the OHC time series show clear quasi periodic variation. Could you compute the periodogram and discuss the observed quasi periodicity?

**Response**: We feel it is beyond the scope of this manuscript to explain the source of these periodic fluctuations in dOHC/dt. They are seen in all three time series (and in previous studies looking at OHU). This is more of a scientific investigation than a data description, which is the intent of this manuscript and why it was submitted to *Earth System Science Data*.